# StepORLM: A Self-Evolving Framework With Generative Process Supervision For Operations Research Language Models

**Chenyu Zhou, Tianyi Xu, Jianghao Lin,**\* **Dongdong Ge**
Antai College of Economics and Management,
Shanghai Jiao Tong University
Shanghai, China
{chenyuzhou, crimsonflag, linjianghao, ddge}@sjtu.edu.cn

## Abstract

Large Language Models (LLMs) have shown promising capabilities for solving Operations Research (OR) problems. While reinforcement learning serves as a powerful paradigm for LLM training on OR problems, existing works generally face two key limitations. First, outcome reward suffers from the *credit assignment problem*, where correct final answers can reinforce flawed reasoning. Second, conventional discriminative process supervision is *myopic*, failing to evaluate the interdependent steps of OR modeling holistically. To this end, we introduce `StepORLM`, a novel self-evolving framework with generative process supervision. At its core, `StepORLM` features a co-evolutionary loop where a policy model and a generative process reward model (GenPRM) iteratively improve on each other. This loop is driven by a dual-feedback mechanism: definitive, outcome-based verification from an external solver, and nuanced, holistic process evaluation from the GenPRM. The combined signal is used to align the policy via Weighted Direct Preference Optimization (W-DPO) and simultaneously refine the GenPRM. Our resulting 8B-parameter `StepORLM` establishes a new state-of-the-art across six benchmarks, significantly outperforming vastly larger generalist models, agentic methods, and specialized baselines. Moreover, the co-evolved GenPRM is able to act as a powerful and universally applicable process verifier, substantially boosting the inference scaling performance of both our own model and other existing LLMs. We release our models and code to facilitate future research[1].

## 1 Introduction

Large language models (LLMs) have shown remarkable capabilities for complex reasoning, enabling new frontiers in specialized domains like Operations Research (OR) (Xiao et al., 2025). Research on applying LLMs to OR has largely advanced along two fronts: 1) developing agentic systems (AhmadiTeshnizi et al., 2024a; Xiao et al., 2024) and 2) improving LLMs through specialized training (Huang et al., 2025). Within the latter, reinforcement learning (RL) has become a dominant paradigm through outcome reward (Chen et al., 2025b) or process supervision (Astorga et al., 2024; Chen et al., 2025a).

However, these established RL paradigms exhibit critical flaws when applied to operations research, which requires long-horizon, interdependent reasoning. Outcome-based supervision, which rewards a model based solely on the final solution's correctness, suffers from the **credit assignment problem**. As illustrated in Figure 1(a), an optimal outcome can emerge from a flawed reasoning process, inadvertently reinforcing incorrect intermediate steps. While process supervision aims to resolve this by rewarding intermediate steps, standard **discriminative step-wise** PRMs often suffer from a **myopic** view, evaluating steps in isolation. This is particularly detrimental in OR, where the validity

---

\*Corresponding author.
[1] https://github.com/0xzhouchenyu/StepORLM

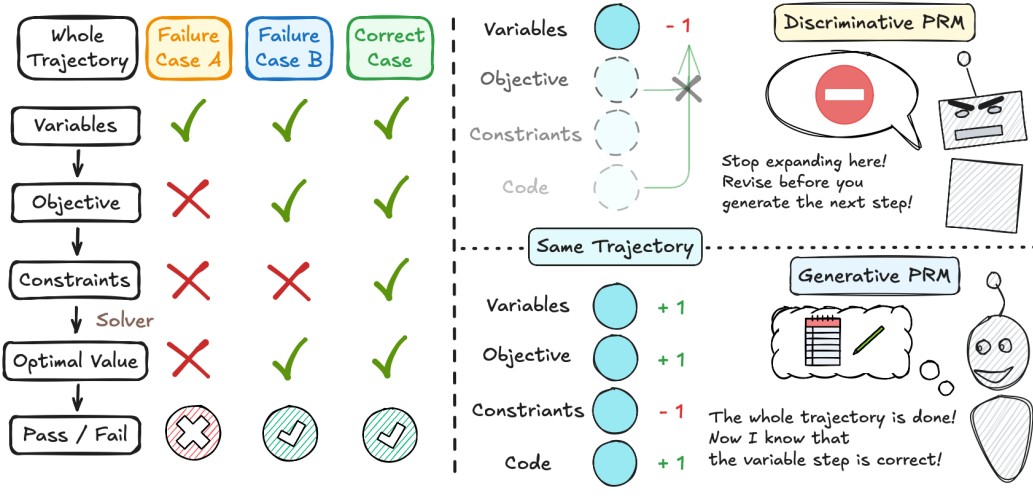

Figure 1: The illustration of (a) the credit assignment problem of outcome reward and (b) the myopic issue of discriminative process supervision, which motivates the application of generative process supervision for operations research language models.

of a constraint depends heavily on variable definitions introduced many steps earlier, making isolated scoring unreliable.

As shown in Figure 1(b), the correctness of an early step, like a variable definition, may only become apparent once subsequent constraints are defined, causing discriminative PRMs to provide unreliable or inconsistent rewards.

Therefore, we argue that the contextual and interdependent nature of OR modeling demands a paradigm shift from myopic, step-wise evaluation to **holistic, trajectory-level process supervision**. The critic should assess the entire reasoning path retrospectively and understand the intricate dependencies between steps before assigning credit. As illustrated in Figure 1(b), this motivates the application of *generative* PRMs that can reason about the completed trajectory and generate a holistic critique, rather than a discriminative one that provides disconnected, localized scores.

To this end, we propose a novel Self-Evolving Framework with Generative Process Supervision for Operations Research Language Models (`StepORLM`). At the heart of our framework is **a co-evolutionary loop** where a policy model and a novel generative process reward model (GenPRM) iteratively improve each other. At each iteration, the policy generates reasoning trajectories that are evaluated from two complementary perspectives: definitive outcome verification from an external solver and nuanced, holistic process feedback from GenPRM. This dual-feedback signal is distilled into 1) preference pairs to align the policy via a Weighted Direct Preference Optimization (W-DPO) objective and 2) filtered trajectory rewarding data to further fine-tune GenPRM. In this way, both the policy model and GenPRM would be iteratively improved through the self-evolving loop, ultimately leading to process-sound and outcome-correct reasoning capabilities to solve complex OR problems.

Our contributions are summarized as follows:

- To the best of our knowledge, we are the first to deploy **generative process supervision** in the **solver-based** OR domain. Unlike general reasoning tasks, we ground the supervision in strict solver feasibility and numerical optimality, mitigating the *credit assignment problem* and *myopia issue* inherent to existing methods.

- We propose `StepORLM`, a novel self-evolving framework centered on the co-evolution of a policy model and a generative process reward model (GenPRM). It integrates a scalable data synthesis pipeline with a dual-feedback loop that grounds the models in both process-level logical consistency and final-outcome correctness.

- Extensive experiments on six benchmarks show that `StepORLM` achieves the **SOTA** performance against both vastly larger generalist/agentic models (e.g., `GPT-4o`) and specialized baselines. We also demonstrate that the co-evolved GenPRM acts as a **universal inference-time verifier**, substantially boosting the performance of other OR language models.

- We release our code, the StepORLM model weights, and the GenPRM verifier weights to the community, providing a strong foundation for future research in LLM-based OR problem solving.

## 2 RELATED WORK

**Large Language Models for Optimization Modeling.** Early works explore using LLMs to parse natural language descriptions into mathematical models. For example, the NL4Opt competition (Ramamonjison et al., 2023) demonstrates that ChatGPT and other LLMs can identify problem entities and generate LP formulations from word problems. Pipeline frameworks (Huang et al., 2025) combine LLM components with traditional solvers: AutoFormulation (Huang et al., 2025; Jiang et al., 2025; Lu et al., 2025; Zhou et al., 2025) uses LLMs to tag variables and constraints in text descriptions, and systems like OptiMUS(AhmadiTeshnizi et al., 2024b) pipeline textual inputs through LLM-to-code translators before invoking MILP solvers. These methods improve accessibility, but often produce hallucinated constraints or missing logic, as the LLMs have no built-in notion of optimization semantics. Agentic methods Zhang et al. (2024); Yang et al. (2025b) are also proposed, which assign specialized LLM agents under a conductor that orchestrates and reflects on partial solutions (Xiao et al., 2024; Deng et al., 2024; AhmadiTeshnizi et al., 2024a). While these approaches outperform simpler pipelines on benchmarks, they still rely on repeated LLM exchanges without explicit verification of each reasoning step. In practice, even advanced models exhibit unsound reasoning: for instance, Jiang et al. (2025) note that alignment and "self-correction" mechanisms are needed to prevent LLM hallucinations in optimization contexts. Concurrent works also explore iterative data synthesis to improve OR reasoning. For instance, Step-Opt (Wu et al., 2025) and Lima et al. (2025) propose pipelines to generate verifiable synthetic data using rule-based checkers or symbolic representations. While these methods focus on enhancing data quality on the data side, StepORLM focuses on the model side by establishing a co-evolutionary loop between the policy and a generative process reward model.

**Advanced Training and Refinement Paradigms.** To enhance the reliability of LLM reasoning, recent work moves beyond simple pipelines to incorporate advanced training and refinement paradigms Fang et al. (2025); Xi et al. (2025). One major direction is reinforcement learning, which is applied with varying feedback granularity. Outcome-driven methods like Solver-Informed RL (SIRL) (Chen et al., 2025b) use a solver's final verdict for precise rewards but suffer from the credit assignment problem. In contrast, standard **discriminative** Process Reward Models (PRMs) provide denser feedback but can be **myopic**, rewarding locally correct steps that are globally inconsistent (Dai et al., 2025; Zheng et al., 2025a).

Very recently, generative process supervision has emerged to address these limitations. For instance, GenPRM (Zhao et al., 2025) and R-PRM (She et al., 2025) utilize generative reasoning to produce holistic critiques rather than isolated scores. Similarly, frameworks like rStar-Math (Guan et al., 2025) demonstrate the power of self-evolution strategies for policy improvement, while others explore inference-time scaling for generalist reward models (Liu et al., 2025). However, these approaches typically rely on answer-string supervision. StepORLM adapts these concepts to the solver-based OR domain, introducing a dual-feedback W-DPO objective and explicitly co-evolving the policy and verifier grounded in numerical feasibility. Complementing these training methods are iterative refinement and search strategies. These techniques embed LLMs into structured search frameworks like beam search (Wang et al., 2024) and Monte Carlo Tree Search (Zheng et al., 2025b; Liu et al., 2024), or employ evolutionary loops with self-reflection to iteratively improve generated solutions (Ye et al., 2024). While these approaches make final outputs more reliable by exploring a wider solution space, they often operate at the output level rather than being deeply integrated into the model's core training process. Thus, a trade-off remains: outcome-based RL lacks intermediate supervision, stepwise PRMs lack global coherence, and search methods are often applied post-hoc. To bridge this gap, discriminative methods based on both final outcome correctness and stepwise feedback can penalize incorrect reasoning steps while maintaining global coherence, effectively combining the strengths of both process- and outcome-based approaches (Chen et al., 2025a).

The research community actively builds end-to-end OR modeling systems that merge symbolic rigor with learned reasoning. This effort is supported by a growing number of benchmarks (Wang et al., 2024; Huang et al., 2024; 2025; Yang et al., 2024; Xiao et al., 2025). StepORLM is designed to address these gaps by embedding process-level supervision and iterative refinement, ensuring each

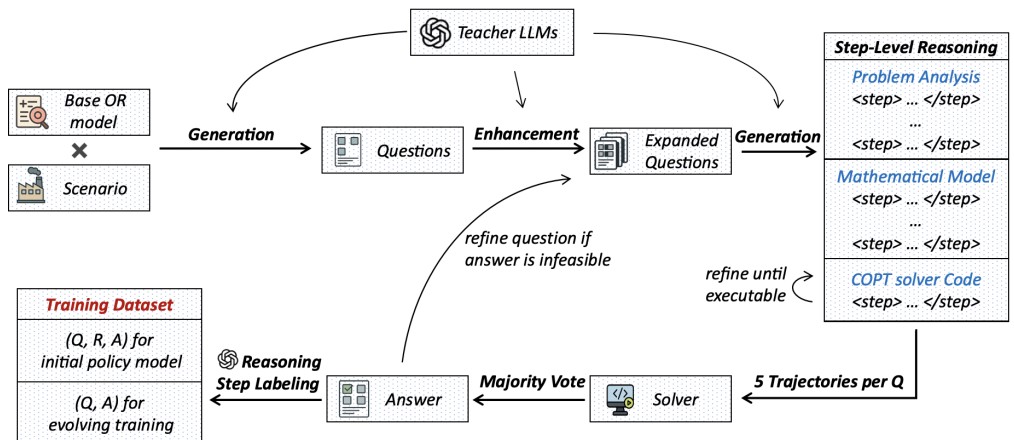

Figure 2: The illustration of data synthesis pipeline for the warm-up stage. A teacher LLM generates diverse OR problems and their step-by-step reasoning solutions. Each solution is rigorously validated and filtered by an external solver, with an automated refinement loop to ensure the quality. This verified corpus is then used to train the initial policy model via SFT.

modeling step is verified. Its co-evolutionary framework combines a generative PRM for nuanced step-level feedback with final-solver rewards, mitigating the credit assignment problem via a weighted DPO objective. By having its policy and generative PRM co-evolve, StepORLM learns to self-verify internally, leading to more robust and error-resistant modeling behavior that advances the goal of reliable and scalable OR reasoning with LLMs.

## 3 METHODOLOGY

### 3.1 OVERVIEW OF STEPORLM

We train `StepORLM` using a two-stage, self-evolving framework designed to instill robust, process-sound reasoning for operations research problems. As illustrated in Figure 2 and Figure 3, the framework is composed of two main stages:

- **Stage 1: Warm-Up for Initial Policy**. As shown in Figure 2, we bootstrap the training process by creating a high-quality, solver-verified dataset, which consists of OR problems and step-by-step solutions. This corpus is used to train an initial policy $\pi_0$ via Supervised Fine-Tuning (SFT). In practice, this warm-up corpus contains 50K verified training instances that cover both synthesized OR modeling tasks and instruction-style OR queries.

- **Stage 2: Iterative Co-Evolution for Policy and GenPRM**. As shown in Figure 3, starting with the initial policy $\pi_0$ and a base LLM as the critic $\rho_{\theta_0}$, we commence a self-evolving training loop. The policy model and the generative process reward model (GenPRM) co-evolve, with the policy generating increasingly better solutions and the GenPRM becoming a more discerning critic. Concretely, we construct a hard subset $\mathcal{D}_{\text{hard}}$ of 13K problems from the 50K warm-up pool, consisting of instances where the warm-up policy with 5-way majority voting fails to reach unanimous correctness. All self-evolving iterations in Stage 2 operate on this hard subset.

### 3.2 WARM-UP FOR INITIAL POLICY

The goal of the warm-up stage is to produce a strong initial policy model capable of generating structured, multi-step solutions to OR problems. We achieve this through a scalable data synthesis pipeline, shown in Figure 2.

We begin with a large pool of base OR problem templates and diverse industry scenarios. A powerful teacher LLM (GPT-4o in this paper) first pairs compatible templates and scenarios to draft a natural-language question $Q$. To enrich the data, each question is augmented through paraphrasing, unit conversion, and controlled parameter scaling. The teacher model then generates a complete, multi-

step reasoning trajectory for each question $Q$, mirroring the workflow of a human expert from problem analysis to executable solver code. See Appendix E for more details.

To ensure data quality, every generated solution undergoes a deterministic validation process: we execute the code, inspect the solver status, and verify the objective value. Trajectories that produce errors would trigger an automatic self-refinement loop, prompting the teacher LLM to revise the formulation or code until validation succeeds or the retry budget is exhausted. For questions with multiple valid solutions, we select the one achieving the best objective value. This rigorous process yields a high-quality dataset of verified $(Q, R)$ pairs, i.e., Question (Q) and Reasoning (R). We use the constructed dataset to train the initial policy model $\pi_0$ via Supervised Fine-Tuning (SFT).

### 3.3 ITERATIVE CO-EVOLUTION FOR POLICY AND GENPRM

Following the warm-up stage, the iterative co-evolution stage begins. As illustrated in Figure 3, this self-evolving loop simultaneously enhances the policy model $\pi_\theta$ and the generative process reward model (GenPRM) $\rho_\phi$. Each iteration involves three key steps: 1) trajectory collection and evaluation, 2) policy alignment, and 3) GenPRM refinement.

#### 3.3.1 TRAJECTORY COLLECTION AND DUAL-SOURCE EVALUATION

In the self-evolving stage, we utilize the hard subset $\mathcal{D}_{hard}$ (defined in Section 3.1) as the prompt source. Crucially, unlike the static training in the warm-up stage, the training data for W-DPO is dynamically generated. At each iteration, the current policy $\pi_\theta$ generates a new set of $k$ candidate solution trajectories for each question in $\mathcal{D}_{hard}$. These trajectories are then put through a dual-source evaluation:

- **Outcome Verification**. The final code of each trajectory is executed, and an external solver provides a definitive, ground-truth label of success or failure.
- **Process Evaluation**. The GenPRM $\rho_\theta$ holistically assesses each complete trajectory and provides step-by-step correctness scores, capturing the quality of the reasoning process. Note that, different from conventional discriminative process supervision, our GenPRM is required to first perform chain-of-thought reasoning over the OR solution and then generate a holistic, post-hoc evaluation of the entire reasoning trajectory. This enables GenPRM to capture long-term dependencies that step-wise verifiers often miss (see Appendix C.2 for a case study on TSP illustrating how this global view prevents reward hacking).

After assessing the generated trajectories, we use the dual-feedback signals to further refine both the policy and GenPRM. The improved models would then start another round of self-evolving iteration. Next, we introduce the policy alignment and GenPRM refinement at each iteration.

#### 3.3.2 POLICY ALIGNMENT VIA W-DPO

The dual-source feedback is first used to align the policy model. We distill the trajectory evaluation results into preference pairs $(\tau_w, \tau_l, w)$, where $\tau_w$ is the winning trajectory and $\tau_l$ is the losing one. The selection logic, as detailed in Algorithm 1, prioritizes solver-verified solutions. If solver outcomes are identical for a pair of trajectories (e.g., both pass or both fail), then the trajectory with a higher ratio of correct steps, as judged by the GenPRM, is preferred.

Next, we design a Weighted Direct Preference Optimization (W-DPO) objective for policy training. The weight $w$ is defined to quantify the quality gap between two trajectories: solver-distinguished pairs receive a fixed, high weight (i.e., $w = 1.0$), while others are weighted proportionally to the difference in their process quality scores $r(\tau)$, i.e., $w = |r(\tau_{pos}) - r(\tau_{neg})|$. Unlike unstable step-wise DPO objectives, this scalar-weighted formulation stabilizes training by aggregating fine-grained process feedback into a robust trajectory-level signal. The final objective focuses the model training on the most informative pairs weighted by $w$:

$$\mathcal{L}_{\text{W-DPO}}(\theta) = -\mathbb{E}_{(x,\tau_w,\tau_l)}\Big[w_{(\tau_w,\tau_l)} \cdot \log \sigma\big(\beta\big(\log \pi_\theta(\tau_w \mid x) - \log \pi_\theta(\tau_l \mid x)\big)\big)\Big], \quad (1)$$

where $w_{(\tau_w,\tau_l)}$ is the weight for trajectory pair $(\tau_w, \tau_l)$, $\pi_\theta(\tau \mid x)$ is the log-likelihood of generating trajectory $\tau$ given the input question $x$, and $\beta$ is a scaling factor.

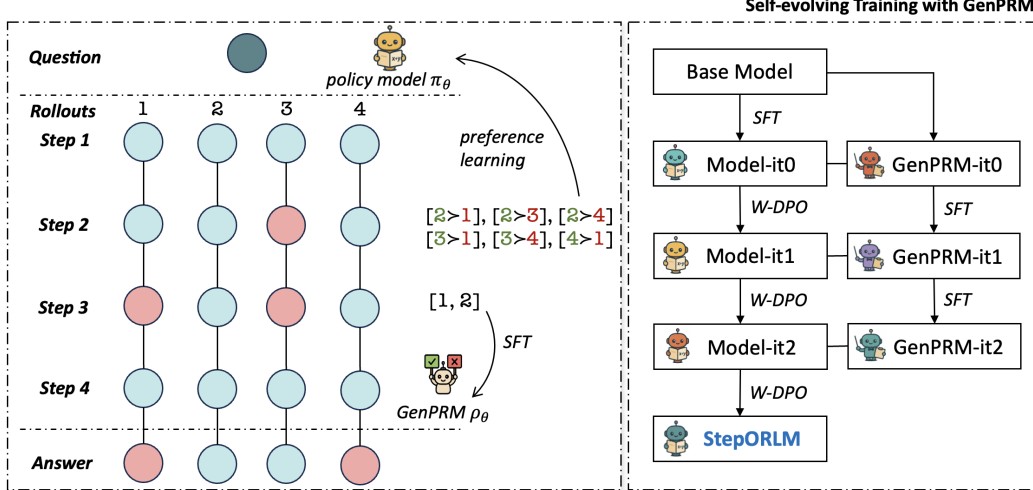

Figure 3: The co-evolutionary loop of StepORLM. At each iteration, the policy model $\pi_\theta$ generates multiple trajectories. The feedback from both the external solver (outcome) and the GenPRM $\rho_\theta$ (process) is used to create training data that simultaneously refines the policy via W-DPO and improves the GenPRM via SFT, fostering reciprocal improvement.

---

**Algorithm 1** Preference Pair Construction

1: **Input:** Two trajectories $\tau_a, \tau_b$.
2: **Output:** A tuple $(\tau_w, \tau_l, w)$.
3: *// Rule 1: A solver-verified trajectory is always preferred.*
4: **if** $\tau_a$ has a correct result and $\tau_b$ does not **then**
5:     **return** $(\tau_a, \tau_b, 1.0)$
6: **else if** $\tau_b$ has a correct result and $\tau_a$ does not **then**
7:     **return** $(\tau_b, \tau_a, 1.0)$
8: **end if**
9: *// Rule 2: Otherwise, prefer the trajectory with a higher ratio of correct steps.*
10: Let diff $\leftarrow$ CorrectStepRatio$(\tau_a)$ − CorrectStepRatio$(\tau_b)$
11: **if** diff $> 0$ **then**
12:     **return** $(\tau_a, \tau_b, \text{diff})$
13: **else if** diff $< 0$ **then**
14:     **return** $(\tau_b, \tau_a, -\text{diff})$
15: **else**
16:     **return** None
17: **end if**

---

### 3.3.3 GENPRM REFINEMENT VIA SFT

The same trajectory evaluation data from the current iteration is then used to refine the critic, i.e., GenPRM. Specifically, to ensure high-quality supervision, we select only solver-consistent trajectories, i.e., instances where GenPRM's verdict aligns with the external solver's outcome. For these high-confidence samples, we treat GenPRM's own generated critique and step-wise judgments as targets to form a new SFT dataset. We provide the exact prompt-completion format and training data schema in Appendix F. This dataset is used to continue the supervised fine-tuning of GenPRM $\rho_\theta$ with process supervision signals.

We can see that such a process creates a positive feedback loop: as the policy improves, it generates more diverse and accurate reasoning paths, providing higher-quality training data for the GenPRM. In turn, a more capable GenPRM provides a more precise and reliable reward signal, further accelerating the policy alignment. This symbiotic refinement is crucial for surpassing the performance of the initial SFT model, which is thus referred to as the self-evolving framework.

# 4 EXPERIMENTS

## 4.1 EXPERIMENT SETUP

**Benchmarks.** We conduct our evaluation on a diverse set of six public benchmarks widely used in the OR community: NL4Opt (Ramamonjison et al., 2023), MAMO (split into EasyLP and ComplexLP) (Huang et al., 2024), NLP4LP (AhmadiTeshnizi et al., 2024b), ComplexOR (Xiao et al., 2024), IndustryOR (Huang et al., 2025) and ReSocratic (Yang et al., 2025c). More details about the six benchmarks are provided in appendix B.

**Baselines.** To ensure a thorough comparison, we benchmark against a comprehensive set of baselines in three categories:

- **Zero-shot Generalist LLMs:** Powerful, large-scale models without specific fine-tuning, including `GPT-4o` (OpenAI, 2024), `OpenAI o3`(OpenAI, 2025), `Gemini-2.5-pro`(Comanici et al., 2025), `Kimi-K2`(Team et al., 2025), `DeepSeek-R1`(Guo et al., 2025), `DeepSeek-V3` (DeepSeek-AI et al., 2025), `Qwen3-32B` (Yang et al., 2025a), `Qwen3-8B`(Yang et al., 2025a), and `Qwen2.5-72B-Instruct` (Qwen et al., 2025).

- **Specialized Fine-tuned LLMs:** Models specifically trained on OR solving tasks, including `ORLM` (8B) (Huang et al., 2025), `LLMOPT` (14B) (Jiang et al., 2025) and `OptMATH` (32B) (Lu et al., 2025). We try our best to re-evaluate all publicly accessible models. For those not open-sourced, we report their original performance according to corresponding published papers. Note that due to our use of a cleaned dataset (Xiao et al., 2025), these original scores serve as a reference and may not be directly comparable.

- **Agentic Methods:** Multi-step reasoning frameworks that orchestrate LLMs to solve problems, including `OptiMUS-v0.3` (AhmadiTeshnizi et al., 2024a), Chain-of-Thought (`CoT`) (Wei et al., 2022), Chain-of-Experts (`CoE`) (Xiao et al., 2023), and `CAFA` (Deng et al., 2024). To ensure a fair and powerful comparison, all agentic methods are implemented using `GPT-4o` as their backbone model.

**Implementation Details** We develop our `StepORLM` models by fine-tuning the publicly available Qwen3-8B (Yang et al., 2025a). Training and inference are conducted on a single node with 8 × NVIDIA H100 80 GB GPUs. We used a synthesized dataset of 50K questions for subsequent training. The model was trained using the AdamW optimizer with a learning rate of 7e-6 and weight decay of 0.1. The maximum sequence length was set to 8192, and we introduced special tokens `<step>` and `</step>` to enhance reasoning. A batch size of 64 was used, and training proceeded for 3 epochs. During the self-evolving sampling phase, we generated 4 trajectories for each response to be used for subsequent preference learning. For W-DPO, we set the hyperparameter $\beta$ to 0.1. To ensure optimal performance and a fair comparison across different frameworks, we selected solvers best suited to each category of model. For general-purpose LLMs and all agentic methods, we used Gurobi due to its broad compatibility. For the fine-tuned models, we used their respectively adapted solvers: both our `StepORLM` and `ORLM` employed the COPT solver, whereas `LLMOPT` is evaluated using Pyomo.

## 4.2 MAIN RESULTS

We report the main results in Table 1. Our `StepORLM` establishes a new state-of-the-art in OR problem-solving tasks, significantly outperforming both vastly larger generalist/agentic models and specialized LLMs. As an 8B-parameter model, `StepORLM` outperforms much larger, state-of-the-art generalist models, such as `DeepSeek-V3` (671B) and `Qwen2.5-72B-Instruct` (72B), and even those agentic methods based on `GPT-4o`. This result underscores the profound effectiveness of specialized, process-supervised training, enabling a compact model to achieve more reliable and accurate OR modeling capabilities than its far larger counterparts.

Moreover, leveraging the co-evolved GenPRM as an inference-time process verifier (i.e., `StepORLM+GenPRM`) further pushes the performance to an average Pass@1 accuracy of 85.6%. Such a configuration excels especially on the most challenging benchmarks, with remarkable accuracy gains of 9.9% on ComplexOR and 9.5% on IndustryOR compared with the basic `StepORLM`. This highlights the dual benefit of our co-evolutionary framework: the policy model learns to generate

Table 1: The overall performance of `StepORLM` and baselines with Pass@1 accuracy (%) on six OR benchmarks. All agentic method baselines utilize `GPT-4o` as the base model. `StepORLM+GenPRM` indicates using the GenPRM as process verifier to enable the inference scaling of `StepORLM`. Scores cited from original publications are marked with the symbol (*), while missing entries are denoted with (-). Best results are highlighted in **bold** and the second-highest values are underlined.

| Model | Params | NL4OPT | MAMO | | NLP4LP | CompOR | IndOR | ReSocratic | Avg. |
|---|---|---|---|---|---|---|---|---|---|
| | | | EasyLP | ComplexLP | | | | | |
| *Zero-shot LLMs* | | | | | | | | | |
| OpenAI o3 | Closed | 78.4 | 93.9 | **63.1** | 93.8 | **72.2** | 76.2 | 84.4 | **80.3** |
| Gemini-2.5-Pro | Closed | **82.6** | 87.9 | 52.3 | **94.9** | 66.7 | **78.6** | **89.6** | 78.9 |
| GPT-4o | Closed | 61.2 | 70.3 | 57.7 | 73.6 | 42.9 | 38.1 | 48.4 | 56.0 |
| Kimi-K2 | 1T | 77.9 | 93.4 | 55.9 | 84.3 | 61.1 | 59.5 | 81.9 | 73.4 |
| DeepSeek-R1 | 671B | 77.5 | 90.3 | 59.5 | 87.6 | 66.7 | 59.5 | 83.9 | 75.0 |
| DeepSeek-V3 | 671B | 79.8 | 95.2 | 53.2 | 92.1 | 55.6 | 66.7 | 85.1 | 75.4 |
| Qwen2.5-72B-Inst | 72B | 78.9 | **95.8** | 44.1 | 88.2 | 50.0 | 57.1 | 81.1 | 70.7 |
| Qwen3-32B | 32B | 77.5 | 92.3 | 46.9 | 93.8 | 50.0 | 61.9 | 85.1 | 72.5 |
| Qwen3-8B | 8B | 63.8 | 73.6 | 45.0 | 58.4 | 27.8 | 52.4 | 61.0 | 54.6 |
| *Fine-tuned LLMs* | | | | | | | | | |
| ORLM | 8B | 73.8 | 90.4 | 59.5 | 76.4 | 50.0 | 42.9 | 61.8 | 65.0 |
| LLMOPT | 14B | 75.1 | 83.5 | 67.6 | 86.0 | 22.2 | 52.4 | 73.2 | 65.7 |
| OptMATH (origin) | 32B | 95.9* | 89.9* | 54.1* | - | - | - | - | - |
| StepORLM | 8B | **97.7** | **97.2** | **79.3** | **97.8** | **55.6** | **59.5** | **82.6** | **81.4** |
| *Agentic Methods* | | | | | | | | | |
| OptiMUS-v0.3 | Closed | 76.2 | 78.0 | 46.8 | 88.8 | 46.8 | 45.2 | 87.6 | 67.1 |
| CoT | Closed | 62.2 | 49.5 | 42.3 | 74.7 | 39.2 | 40.5 | 43.6 | 50.3 |
| CoE | Closed | 66.7 | 94.4 | 50.6 | 87.4 | 57.1 | 31.2 | 71.2 | 65.5 |
| CAFA | Closed | 68.1 | 71.2 | 44.5 | 50.0 | 46.4 | 41.1 | 40.1 | 51.6 |
| StepORLM+GenPRM | 8B+8B | **97.2** | **97.8** | **87.4** | **98.9** | **61.1** | **61.9** | **94.6** | **85.6** |

**Abbreviations**: CompOR: ComplexOR, IndOR: IndustryOR, Avg: Macro-Average, Qwen2.5-72B-Inst: Qwen2.5-72B-Instruct.

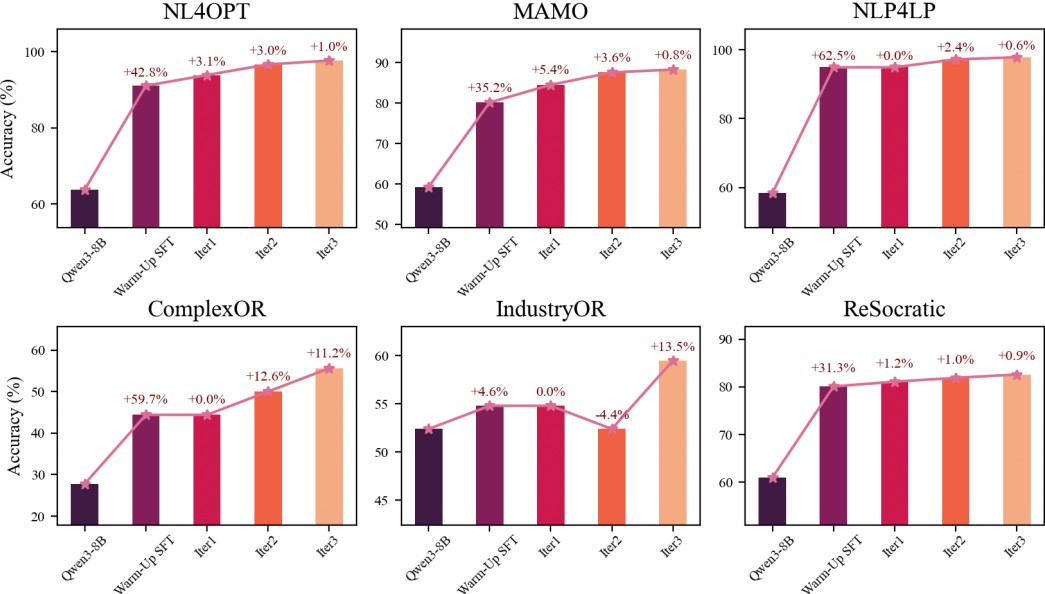

Figure 4: The analysis on the self-evolving process by tracking the performance (Pass@1 accuracy) at each training iteration. The relative improvement of current iteration over the previous one is demonstrated on the corresponding bar.

high-quality solutions, while the GenPRM learns to effectively identify and select the best one. Furthermore, StepORLM excels at avoiding reward hacking, a point we illustrate with a case study in Appendix C.1.

Table 2: Performance comparison of `StepORLM` under different inference scaling strategies.

| Model | NL4OPT | MAMO | | NLP4LP | CompOR | IndOR | ReSocratic | Avg. |
|---|---|---|---|---|---|---|---|---|
| | | EasyLP | ComplexLP | | | | | |
| **StepORLM as Policy Model** | | | | | | | | |
| StepORLM | 97.7 | 97.2 | 79.3 | 97.8 | 55.6 | 59.5 | 82.6 | 81.4 |
| + Major Vote | 97.2 | 97.6 | 81.1 | 96.6 | **61.1** | 61.9 | 89.3 | 83.5 |
| + Solver Exec | 97.7 | **98.4** | 81.1 | 96.1 | **61.1** | **66.7** | 90.3 | 84.5 |
| + Discriminative PRM | 97.2 | 97.2 | 81.1 | 97.2 | 55.6 | 59.5 | 87.8 | 82.2 |
| + GenPRM (initial) | **97.8** | 97.6 | 82.8 | 97.2 | 55.6 | 58.5 | 93.1 | 83.2 |
| + GenPRM (final) | 97.2 | 97.8 | **87.4** | **98.9** | **61.1** | 61.9 | **94.6** | **85.6** |
| **ORLM as Policy Model** | | | | | | | | |
| ORLM | 73.8 | 90.4 | 59.5 | 76.4 | **50.0** | 42.9 | 61.8 | 65.0 |
| + Major Vote | 78.7 | 88.4 | 50.5 | 78.7 | 44.4 | 47.6 | 73.0 | 65.9 |
| + Solver Exec | 82.2 | 88.6 | 63.1 | 79.8 | 44.4 | 52.4 | 78.9 | 69.9 |
| + Discriminative PRM | 75.1 | **91.7** | 63.1 | 82.0 | **50.0** | 54.8 | 74.7 | 70.2 |
| + GenPRM (initial) | 87.3 | 90.6 | 55.0 | 90.4 | 44.4 | 47.6 | 65.5 | 68.7 |
| + GenPRM (final) | **91.5** | 91.0 | **64.9** | **91.0** | **50.0** | **57.1** | 79.4 | **75.0** |

## 4.3 ANALYSIS OF SELF-EVOLVING PROCESS

We conduct in-depth analysis on the self-evolving process by tracking the performance at each training iteration, including the warm-up and co-evolving stages. The results are reported in Figure 4, from which we can obtain the following observations:

- The warm-up supervised fine-tuning (SFT) provides a massive foundational performance lift across all benchmarks (e.g., 62.5% accuracy improvement on NLP4LP).

- The subsequent self-evolution iterations consistently build upon the prior models, delivering incremental but crucial accuracy improvements. This confirms that the model progressively refines its capabilities through our self-evolving loop, rather than stagnating after initial fine-tuning.

- The performance trend is notably non-monotonic on the IndustryOR dataset, where the performance first declines and finally largely increases at the third iteration. We attribute this to the small size of testing set and the challenging questions in IndustryOR. Our case-level inspection reveals a potential progression. Early iterations primarily rectify structural modeling errors flagged by the PRM, whereas later iterations concentrate on code-level issues (e.g., index-out-of-bounds errors), which finally leads to accuracy improvements.

## 4.4 INFERENCE SCALING WITH GENPRM

In this section, we investigate the generalization capability of our co-evolved GenPRM as a process verifier for inference scaling. To evaluate this, we employ a Best-of-$k$ strategy with a fixed sampling budget of $k = 4$. Specifically, we sequentially sample candidate trajectories and select the first one that GenPRM verifies as process-correct; if no trajectory passes verification within the budget, the final sample is selected. We apply this method to our final `StepORLM` and another open-sourced `ORLM` baseline, comparing its effectiveness against other inference scaling strategies including majority voting, solver execution, and discriminative PRM.

As shown in Table 2, our self-evolved GenPRM is generally the most effective inference scaling strategy, achieving the highest accuracy on average for both `StepORLM` and `ORLM`. When paired with `ORLM`, our GenPRM boosts its average performance by 10%, demonstrating its powerful generalization. It learns fundamental, model-agnostic principles of valid OR reasoning, allowing it to function as a universal verifier that significantly improves the performance of other models, not just its co-evolved policy.

## 4.5 ABLATION STUDY

To study the impact of each component in our proposed self-evolving framework, we conduct experiments based on the following variants: (1) w/o SFT: skipping the warm-up supervised fine-tuning stage; (2) w/o self-evolution: directly sampling a single reasoning trajectory after SFT and

Table 3: Ablation study of StepORLM on Pass@1 accuracy (%). The best results are given in **bold**, and the second-best values are underlined.

| Model | NL4OPT | MAMO | | NLP4LP | CompOR | IndOR | ReSocratic | Avg. |
|---|---|---|---|---|---|---|---|---|
| | | EasyLP | ComplexLP | | | | | |
| StepORLM | **97.7** | **97.2** | **79.3** | **97.8** | **55.6** | **59.5** | **82.6** | **81.4** |
| w/o SFT | 71.4 | 88.3 | 55.9 | 72.5 | 38.9 | 50.0 | 75.2 | 64.6 |
| w/o Self-evolution | 96.7 | **97.2** | 67.6 | 94.9 | 44.4 | 52.4 | 81.4 | 76.4 |
| w/o GenPRM Evolution | 94.4 | 96.3 | 68.5 | **97.8** | 50.0 | 54.8 | 82.4 | 77.7 |
| w/o W-DPO | 97.2 | 95.8 | 73.9 | 96.1 | 50.0 | 52.4 | 80.6 | 78.0 |

applying DPO without iterative refinement; (3) w/o GenPRM Evolution: using the initial GenPRM with frozen parameters for all subsequent iterations; (4) w/o W-DPO: replacing the W-DPO with standard DPO during the self-evolving training phase. The results are shown in Table 3.

Removing the warm-up SFT (w/o SFT) causes the largest performance drop, confirming the necessity of a strong initial policy. Disabling the iterative loop (w/o Self-evolution) prevents the model from refining its initial capabilities, leading to lower scores on challenging benchmarks like ComplexOR. Freezing the GenPRM after initialization (w/o GenPRM Evolution) also degrades performance, validating the importance of the co-evolutionary dynamic between the policy and the critic. Finally, replacing our W-DPO objective with standard DPO (w/o W-DPO) results in a noticeable performance decrease, indicating that our trajectory-level quality weighting provides a more effective learning signal between different trajectory pairs. Together, these results demonstrate that the synergy among all components drives the success of the StepORLM framework.

To disentangle data quality from data scale, we additionally run a controlled SFT study on LLaMA-3-8B-Instruct where both our warm-up corpus and ORLM's OR-Instruct dataset are sub-sampled to 3K instances. Under identical training settings, the model trained on OR-Instruct reaches an average Pass@1 of $60.9\%$, whereas training on our 3K warm-up subset yields $66.8\%$ (+5.9 points), indicating that the gains from Stage 1 mainly stem from the quality of our curated data rather than simply using more examples (see Appendix D.1 for the full breakdown).

We also instantiate the full StepORLM pipeline on LLaMA-3-8B-Instruct, the same backbone used by ORLM. Starting from a weak base model with only $16.1\%$ average Pass@1, warm-up SFT on our 50K corpus lifts performance to $69.4\%$, three self-evolving iterations further improve it to $82.2\%$, and GenPRM-based inference scaling pushes it to $85.0\%$. On this shared backbone, StepORLM therefore surpasses the released ORLM checkpoint ($65.0\%$ average Pass@1) by more than 17 points, demonstrating that our co-evolving framework is effective and compatible across different policy backbones (see Appendix D.2 for the per-benchmark breakdown).

## 5 CONCLUSION

We design **StepORLM**, a novel self-evolving framework to overcome critical limitations in training large language models for operations research. Our approach addresses the credit assignment problem of outcome-based rewards and the myopia of conventional process supervision through a co-evolutionary loop between a policy model and a generative process reward model (GenPRM). StepORLM integrates definitive solver-based outcome verification with holistic, generative process feedback. Such a dual-feedback mechanism ensures the process-sound and outcome-correct reasoning capabilities of the policy model to solve complex OR problems. Our 8B-parameter StepORLM achieves the SOTA performance across six benchmarks, significantly outperforming much larger generalist models, agentic systems, and specialized baselines. Furthermore, the co-evolved GenPRM is able to act as a powerful, universal inference-time process verifier, substantially enhancing the performance of our model and other LLMs for OR problem-solving tasks.

## ACKNOWLEDGMENTS

Lin's research is supported by the National Natural Science Foundation of China (NSFC) [Grant NSFC-624B2096, 72542012, 72595872]. Ge's research is partially supported by the National Natural Science Foundation of China (NSFC) [Grant NSFC-72225009, 72394360, 72394365].

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

## A    LLM Usage Declaration

In the preparation of this manuscript, a Large Language Model (LLM) was utilized exclusively for the purpose of language editing and refinement. The LLM's role was limited to improving grammar, syntax, and clarity. All scientific content, including the conceptual framework, data analysis, and interpretation of results, is the original work of the authors.

## B    Benchmarks

A mass of datasets and methodologies for dataset construction have been proposed to improve the capacity of LLM to address OR problems, with these datasets becoming increasingly intricate and realistic(Yang et al., 2025c)(AhmadiTeshnizi et al., 2024b)(Yang et al., 2024). A recent survey by (Xiao et al., 2025) provides a comprehensive and timely review of benchmark datasets for optimization problems described in natural language. The study identified prevalent issues within the datasets, including logical fallacies, ambiguous problem definitions and incorrect ground-truth answers, and also corrected a portion of the errors identified. Based on that, our work builds upon their initial efforts by conducting further manual verification and cleaning of these widely-used benchmarks. Subsequently, the data is processed into a unified format, thereby establishing a more robust and standardized foundation for subsequent research and model assessment.

### B.1    NL4Opt

The NL4Opt dataset was introduced by Ramamonjison et al. as part of the NeurIPS 2022 NL4Opt competition (Ramamonjison et al., 2022; 2023). It consists of approximately 1,100 annotated word problems describing linear programming (LP) scenarios, with the primary purpose of bridging natural language descriptions and formal optimization models. In a survey, Xiao et al. reported that NL4Opt has a complexity score of 5.59 and an original error rate of at least 26.4%. Each problem presents a self-contained, real-world scenario (e.g., resource allocation, scheduling) with annotations identifying key optimization elements. The official shared task for the dataset is divided into two subtasks: (1) recognizing and labeling problem entities and (2) generating a corresponding mathematical model. The original dataset provides 713 training, 99 validation, and 289 test instances. For our evaluation, we selected 213 validated problems from this benchmark.

### B.2    MAMO

The MAMO dataset was introduced by Huang et al. (2024) as a benchmark for evaluating large language models on complex mathematical modeling tasks, with a specific focus on the formulation process itself rather than just solution accuracy (Huang et al., 2024). The dataset concentrates on linear programming (LP) and mixed-integer linear programming (MILP) problems, excluding nonlinear or differential equation modeling. MAMO is divided into two main components: Easy LP (652 instances) and Complex LP (211 instances). According to an analysis by Xiao et al., EasyLP has a complexity of 7.12 with an error rate of at least 8.13%, while ComplexLP is significantly more difficult, with a complexity of 13.35 and an error rate of at least 23.7%. By delegating the solving step to an OR solver, MAMO focuses specifically on whether an LLM can accurately translate a detailed natural language description into a correct mathematical formulation. From this benchmark, we selected 545 easy and 111 complex instances for our study.

### B.3    NLP4LP

The NLP4LP benchmark, introduced by AhmadiTeshnizi et al. (2023) in their work on the OptiMUS system (AhmadiTeshnizi et al., 2024b;a), evaluates LLM-based agents on translating natural language operations research problems into solver-ready code and mathematical models. The survey by Xiao et al. assigned it a complexity score of 5.58 and identified an error rate of at least 21.7% in the original dataset. It consists of 269 human-authored LP and MILP problems, where each instance presents an optimization scenario from classical OR domains such as scheduling, knapsack allocation, and production planning. The benchmark provides a fine-grained testbed for assessing formulation accuracy. In our work, we chose to evaluate 178 verified instances from NLP4LP.

### B.4 COMPLEXOR

The ComplexOR dataset was introduced by Xiao et al. as part of their Chain-of-Experts framework to stress-test the reasoning and modeling capabilities of LLMs on scenarios more challenging than those in earlier datasets (Xiao et al., 2024). The analysis by Xiao et al. noted a complexity score of 5.98 and a significant error rate of at least 24.3%. ComplexOR was curated from advanced OR case studies and classic difficult optimization problem classes, such as multi-step lot-sizing problems with setup costs, intricate scheduling tasks, and supply chain design problems. The problems are primarily large-scale or conceptually complex mixed-integer linear programs that require multi-step reasoning and often involve hierarchical or time-indexed structures. The dataset's intended purpose is to facilitate the development and evaluation of advanced reasoning strategies for OR problem solving. Following our verification process, we selected 18 reliable problems from this dataset for evaluation.

### B.5 INDUSTRYOR

The IndustryOR dataset, introduced by Huang et al. (2024), is the first industrial operations research benchmark for large language models (Huang et al., 2025). It comprises 100 real-world optimization scenarios from sectors like manufacturing, logistics, finance, and energy. The study by Xiao et al. highlighted this dataset's high difficulty, reporting a complexity score of 14.06 and a substantial error rate of at least 54.0%. Designed to evaluate LLMs on practical, domain-specific tasks, IndustryOR covers five OR categories—linear, integer, mixed-integer, nonlinear, and others—and is labeled by difficulty (Easy, Medium, Hard). For our experiments, we selected 42 validated problems from this benchmark.

### B.6 RESOCRATIC

The ReSocratic dataset is introduced by (Yang et al., 2025c), the name of which is actually a data synthesis method proposed by this work as well. This method synthesize the formatted optimization demonstration in a reverse manner first, and then back-translates it into a question. Through these intermediate reasoning steps, ReSocratic has a higher quality than previous methods. In this paper, we selected 403 validated problems from the original ReSocratic-29K dataset.

## C CASE STUDIES

### C.1 VISUALIZING THE ALIGNMENT PROBLEM IN TSP

The following presentation will outline a comprehensive prototype of the failure case B, as referenced in Figure 1. The sequence of events in the present case study is as follows: firstly, a sketch map of the problem is presented; secondly, a detailed problem description is given; and thirdly, the responses given by Qwen3 and our model StepORLM are outlined. As previously stated, the ensuing discourse will focus on the results obtained from the two models. The graph illustrates the problem and demonstrates that an error in the constraints step will not affect the optimal value.

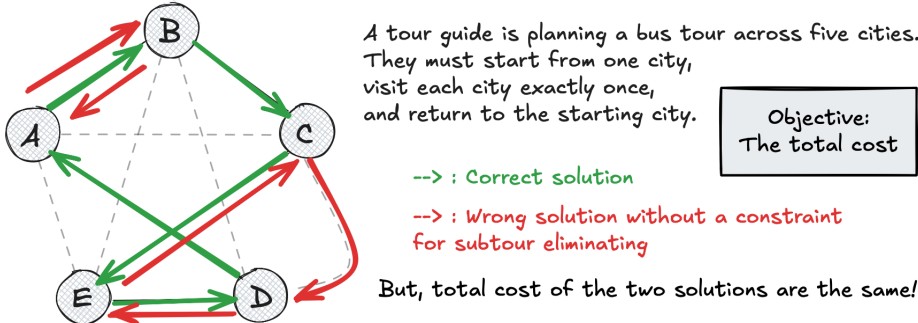

Figure 5: A simplified illustration of problem 74 in ComplexLP

This is a problem with the identifier 74 from the ComplexLP dataset:

---

**Problem Description**

Consider a scenario where a tour guide is planning a bus tour across five cities, named E, F, G, H, and I. The tour must start and end in the same city, and each city should be visited exactly once. The objective is to minimize the total cost of the tour, which could be influenced by factors such as distance, tolls, and fuel expenses.

Here are the travel costs between the cities:

From City E, it costs 37 units to travel to F, 72 units to G, 66 units to H, and 33 units to I. Travelling from City F, the costs are 37 units to E, 26 units to G, 56 units to H, and a whopping 100 units to I. If the journey starts from City G, it takes 72 units to reach E, 26 units to F, 49 units to H, and 32 units to I. From City H, it costs 66 units to get to E, 56 units to F, 49 units to G, and a mere 13 units to I. Lastly, from City I, the travel costs are 33 units to E, 100 units to F, 32 units to G, and 13 units to H.

Keeping these costs in mind, what is the minimum total travel cost for the tour guide to take the bus to each city exactly once and return to the starting city?

---

And the following is our generate prompt for both Qwen3 and StepORLM.

---

**Generate Prompt**

You are a helpful assistant with expertise in mathematical modeling, Python code and the COPT solver. When the User provides an optimization question, you will analyze it, build a detailed mathematical model, and provide the COPT code to solve it.

Your response should follow these steps:
1. Problem Description
2. Sets and Parameters
3. Decision Variables
4. Objective Function
5. Constraints
6. Mathematical Model
7. Nonlinear Relationships
8. Final Model
9. Python Code Using coptpy

please enclose the content within <step>...</step>.

Below is an optimization modeling question. Build a mathematical model and corresponding python code using coptpy that appropriately addresses the question:

PROBLEM DESCRIPTION HERE

Keeping these costs in mind, what is the minimum total travel cost for the tour guide to take the bus to each city exactly once and return to the starting city?

* Make sure to import necessary packages, such as 'import coptpy as cp' and 'from coptpy import COPT'.
* When you create a model make sure to use 'env = cp.Envr()' and 'model = env.createModel'
* When you add a variable, use 'vtype = COPT.'
* Do not name variables and constraints
* Use '.addConstr' or '.addConstrs' to add constraints. If you want to set 'lb' or 'ub' as infinity, please use 'lb=COPT.INFINITY' or 'ub=COPT.INFINITY' instead of 'cp.INFINITY'.
* When you set objective, you should use the 'model.setObjective' method and use 'COPT.MINIMIZE' or 'COPT.MAXIMIZE'.

---

* Do not use 'model.optimize()'.
* Make sure to use 'model.solve()' to solve the question.
* The code output statement is:
if model.status == COPT.OPTIMAL: solution = var.getName(): var.X for var in model.getVars() print('Just print the best obj:', model.ObjVal) else: print('No Solution')

Think step by step.

### C.1.1 STEPORLM'S RESPONSE

**Inference**

$< step >$
**Problem Description:**
The tour guide needs to plan a bus tour that visits each of the five cities exactly once and returns to the starting city, minimizing the total travel cost. This problem is a classic example of the Traveling Salesman Problem (TSP), which is a well-known combinatorial optimization problem.
$< /step >$
$< step >$
**Sets and Parameters:**
- **Cities**: E, F, G, H, I
- **Cost Matrix**: A matrix where each element $c_{ij}$ represents the cost of traveling from city $i$ to city $j$.
$< /step >$
$< step >$
**Decision Variables:**
- $x_{ij}$: A binary variable that is 1 if the tour includes a direct path from city $i$ to city $j$, and 0 otherwise.
$< /step >$
$< step >$
**Objective Function:**
Minimize the total travel cost:

$$\text{Minimize} \quad \sum_i \sum_j c_{ij} \cdot x_{ij}$$

$< /step >$
$< step >$
**Constraints:**
1. **Each city is visited exactly once:**

$$\sum_j x_{ij} = 1 \quad \forall i$$

2. **Each city is left exactly once:**

$$\sum_i x_{ij} = 1 \quad \forall j$$

3. **Subtour elimination constraints:**

$$u_i - u_j + n \cdot x_{ij} \leq n - 1 \quad \forall i, j \geq 1, i \neq j$$

where $u_i$ is an auxiliary variable representing the position of city $i$ in the tour, and $n$ is the number of cities.
$< /step >$
$< step >$

**Mathematical Model:**

$$\min \quad \sum_i \sum_j c_{ij} \cdot x_{ij}$$

$$\text{s.t.} \quad \sum_j x_{ij} = 1 \quad \forall i$$

$$\sum_i x_{ij} = 1 \quad \forall j$$

$$u_i - u_j + n \cdot x_{ij} \leq n - 1 \quad \forall i, j \geq 1, i \neq j$$

$$x_{ij} \in \{0, 1\} \quad \forall i, j$$

$< /step >$
$< step >$
**Nonlinear Relationships:**
No nonlinear relationships are present in this model. This step is unnecessary. $< /step >$
$< step >$
**Final Model:**
The final linear programming model is defined by the objective function and constraints above.
$< /step >$

---

**Python Code Using COPT**

```python
import coptpy as cp
from coptpy import COPT

# Define the cost matrix
cost_matrix = [
[0, 37, 72, 66, 33],
[37, 0, 26, 56, 100],
[72, 26, 0, 49, 32],
[66, 56, 49, 0, 13],
[33, 100, 32, 13, 0]
]

# Number of cities
n = len(cost_matrix)

# Create COPT environment and model
env = cp.Envr()
model = env.createModel("BusTourOptimization")

# Create binary decision variables
x = {}
for i in range(n):
for j in range(n):
    if i != j:
        x[i, j] = model.addVar(vtype=COPT.BINARY, name=f"x_{i}_{j}")

# Create auxiliary variables
u = {}
for i in range(1, n):
u[i] = model.addVar(vtype=COPT.CONTINUOUS, name=f"u_{i}")

# Set objective function
obj = cp.quicksum(
cost_matrix[i][j] * x[i, j]
for i in range(n)
for j in range(n)
```

```
37  if i != j
38  )
39  model.setObjective(obj, sense=COPT.MINIMIZE)
40
41  # Add constraints: each city visited once
42  for i in range(n):
43      model.addConstr(cp.quicksum(x[i, j] for j in range(n) if i != j) ==
            1)
44
45  # Add constraints: each city left once
46  for j in range(n):
47      model.addConstr(cp.quicksum(x[i, j] for i in range(n) if i != j) ==
            1)
48
49  # Add subtour elimination constraints
50  for i in range(1, n):
51      for j in range(1, n):
52          if i != j:
53              model.addConstr(u[i] - u[j] + n * x[i, j] <= n - 1)
54
55  # Solve the model
56  model.solve()
```

### C.1.2 QWEN3'S RESPONSE

**Python Code Using COPT**

```
1   import coptpy as cp
2   from coptpy import COPT
3
4   env = cp.Envr()
5   model = env.createModel()
6
7   # Define cost matrix
8   cost = [
9   [0, 37, 72, 66, 33],
10  [37, 0, 26, 56, 100],
11  [72, 26, 0, 49, 32],
12  [66, 56, 49, 0, 13],
13  [33, 100, 32, 13, 0]
14  ]
15
16  # Create variables
17  x = [[None for _ in range(5)] for _ in range(5)]
18  for i in range(5):
19      for j in range(5):
20          if i != j:
21              x[i][j] = model.addVar(vtype=COPT.BINARY)
22
23  # Objective function
24  obj_expr = 0
25  for i in range(5):
26      for j in range(5):
27          if i != j:
28              obj_expr += cost[i][j] * x[i][j]
29  model.setObjective(obj_expr, COPT.MINIMIZE)
30
31  # Constraints
32  for i in range(5):
```

```
33  model.addConstr(sum(x[i][j] for j in range(5) if j != i) == 1)
34  for i in range(5):
35  model.addConstr(sum(x[j][i] for j in range(5) if j != i) == 1)
36
37  # Solve
38  model.solve()
```

### C.1.3 Discussions on two models' results

This example clearly shows why rewards should be based on step-by-step reasoning rather than just the optimal value. During the reasoning process, StepORLM correctly recognises the importance of subtour elimination constraints, whereas Qwen3 initially identifies them but then overlooks them due to circular self-debate. In this case, the presence or absence of these constraints does not affect the final result. Therefore, both codes can obtain the correct optimal solution.

## C.2 GenPRM vs. Discriminative Verification in TSP

In this section, we provide a concrete case study on the Traveling Salesman Problem (TSP) to illustrate how GenPRM avoids the "reward hacking" (or myopia) issues common in standard discriminative PRMs.

### C.2.1 The Challenge: Strategic Consistency vs. Local Correctness

In optimization modeling, a locally valid mathematical decision can sometimes lead to implementation failure steps later. A classic example is the choice of subtour elimination formulation in TSP:

- **DFJ Formulation:** Uses only binary variables $x_{ij}$ but requires exponential subset constraints ($\sum x_{ij} \leq |S| - 1$). This is mathematically elegant but impossible to implement in a static script without complex callbacks or row generation.
- **MTZ Formulation:** Uses binary variables $x_{ij}$ plus auxiliary continuous variables $u_i$ and polynomial constraints. This is less intuitive but easily implementable in standard solvers like COPT or Gurobi.

A **myopic discriminative PRM** evaluating steps in isolation tends to reward the definition of $x_{ij}$-only variables (Step 3) as CORRECT, because it is mathematically valid. However, this leads to a dead end at the coding step (Step 9).

In contrast, as shown in the comparison below, **GenPRM** evaluates the entire trajectory. It detects that the code failure in Step 9 stems from the insufficient variable definition in Step 3. Consequently, it retroactively marks Step 3 as INCORRECT, forcing the policy to adopt the robust MTZ formulation.

### C.2.2 Comparative Trajectories

Below we display the process supervision results for two attempts. Note how GenPRM identifies the "strategic error" in Attempt 1.

---

**Attempt 1: The Myopic Trap (DFJ Formulation)**
*The model chooses a mathematically valid but practically unimplementable strategy.*

---

**Student Trajectory (Excerpt):**

> **Step 3 [Decision Variables]:**
> - $x_{ij}$: Binary variable, equals 1 if the path from city $i$ to city $j$ is included.
> *(Note: Missing auxiliary variables $u_i$)*
> **Step 5 [Constraints]:**
> 1. Assignment constraints...
> 2. Subtour elimination (DFJ): $\sum_{i,j \in S} x_{ij} \leq |S| - 1 \quad \forall S \subset N$

**Step 9 [Python Code]:**

```
...
# Note: Since listing all subsets is computationally expensive...
# I will omit the specific subset constraints for this simple pass.
# This might result in subtours in the solution.
model.solve()
```

## GenPRM Verification Output:

```
STEP_1: CORRECT
STEP_2: CORRECT
```

**STEP_3: INCORRECT**
EXPLANATION_3: While defining only binary variables $x_{ij}$ is theoretically
sufficient for the DFJ formulation, it is strategically incorrect in this
context. As revealed in Step 9, the model fails to implement the necessary
subset constraints because it lacks a row-generation mechanism. To ensure
a solvable and implementable static model, the decision variables should
have included auxiliary continuous variables $u_i$ for the MTZ formulation.

STEP_4: INCORRECT (Cascading error)
STEP_5: INCORRECT (Relies on incomplete variable set)
...
STEP_9: INCORRECT

---

### Attempt 2: The Globally Consistent Solution (MTZ Formulation)
*Guided by GenPRM, the model adopts a strategy that ensures code executability.*

---

## Student Trajectory (Excerpt):

**Step 3 [Decision Variables]:**
- $x_{ij}$: Binary variable...
- $u_i$: Continuous variable, representing the visit order of city $i$.

**Step 5 [Constraints]:**
2. Subtour elimination (MTZ formulation):
$u_i - u_j + |N| \cdot x_{ij} \leq |N| - 1$

**Step 9 [Python Code]:**
```
...
for i in range(1, n):
for j in range(1, n):
if i != j: model.addConstr(u[i] - u[j] + n * x[i, j] <= n - 1)
model.solve()
```

## GenPRM Verification Output:

```
STEP_1: CORRECT
STEP_2: CORRECT
```

**STEP_3: CORRECT**
EXPLANATION_3: The decision variables are correctly defined. Including the
auxiliary variables $u_i$ along with the binary variables $x_{ij}$ is the correct
strategy for implementing the MTZ subtour elimination formulation, which
allows for a direct static implementation in the code.

```
...
STEP_9: CORRECT
EXPLANATION_9: The Python code perfectly implements the MTZ formulation
derived in the previous steps.
```

This TSP case study concretely illustrates the "myopic PRM" failure mode discussed in the main text: a purely step-wise discriminative verifier can reward locally valid but globally unusable modeling choices (DFJ-style subtour elimination), whereas a solver-aware GenPRM jointly reasoning over the whole trajectory steers the policy toward the implementable MTZ formulation. In our full experiments, combining GenPRM with strict solver feedback in the proposed dual-feedback framework prevents such reward hacking from becoming systematic or degrading overall performance.

# D    ADDITIONAL ABLATION STUDIES

## D.1    IMPACT OF DATA QUALITY (CONTROLLED SFT)

To disentangle the impact of data quality from data scale, we compare our warm-up SFT data against the OR-Instruct dataset (Huang et al., 2025). We sample 3K instances from our warm-up data to match the size of the released OR-Instruct set and train a LLaMA-3-8B-Instruct backbone under identical settings. As shown in Table 4, our data yields a +5.9% average improvement, highlighting the superior quality of our synthesized corpus.

Table 4: Controlled SFT performance comparison (3K training samples) on LLaMA-3-8B-Instruct.

| Model | NL4Opt | EasyLP | CompLP | NLP4LP | CompOR | IndOR | ReSoc | Avg |
|---|---|---|---|---|---|---|---|---|
| OR-Instruct (3K) | 80.8 | 85.5 | 50.5 | 80.9 | 27.8 | 31.0 | 69.7 | 60.9 |
| Warm-up (3K) | **89.2** | **90.0** | **55.0** | **87.1** | **33.3** | **40.5** | **72.2** | **66.8** |

## D.2    STEPORLM PIPELINE ON LLAMA-3-8B BACKBONE

To demonstrate the backbone compatibility of our framework, we instantiate the full StepORLM pipeline on LLaMA-3-8B-Instruct. Table 5 details the performance evolution across the warm-up, self-evolution, and inference scaling stages. The consistent gains confirm that our framework is robust across different model architectures.

Table 5: Performance evolution of StepORLM using LLaMA-3-8B-Instruct as the backbone.

| Training Phase | NL4Opt | EasyLP | CompLP | NLP4LP | CompOR | IndOR | ReSoc | Avg |
|---|---|---|---|---|---|---|---|---|
| Base (LLaMA-3-8B) | 19.2 | 18.9 | 14.4 | 18.5 | 11.1 | 7.1 | 23.8 | 16.1 |
| + Warm-up SFT | 88.3 | 86.2 | 69.4 | 87.6 | 27.8 | 47.6 | 79.2 | 69.4 |
| + Self-evol. (StepORLM) | 97.2 | 97.8 | 84.7 | 93.8 | 55.6 | 61.9 | 84.6 | 82.2 |
| + Inference Scaling | **97.7** | **98.0** | **87.4** | **96.6** | **61.1** | **66.7** | **87.3** | **85.0** |

# E    STEP-LEVEL REASONING TRAJECTORY TEMPLATE

Each reasoning step is enclosed within `<step>...</step>` tags. While this structure is fixed during the initial warm-up stage, it becomes more flexible in subsequent self-evolving iterations. This is because we do not constrain the model's output format or the number of steps during this phase, allowing the LLM to learn more complex and implicit reasoning patterns

1. **Problem Description:** Clearly identify the problem, restating unclear expressions and defining the optimization objective.

2. **Sets and Parameters:** Define all relevant sets (e.g., products, time periods, locations) and parameters (e.g., costs, capacities, demands).

3. **Decision Variables:** Introduce appropriate decision variables, specifying their domains and interpretations.

4. **Objective Function:** Formulate the mathematical expression for the objective to be minimized or maximized.

5. **Constraints:** Develop all necessary constraints, explaining the reasoning behind each.

6. **Mathematical Model Summary:** Present a concise mathematical formulation of the complete model.

7. **Nonlinear Relationships (if applicable):** Handle any nonlinear relationships through transformation or approximation techniques.

8. **Final Model and Implementation Considerations:** Articulate the detailed model with any implementation-specific adaptations.

# F   TRAINING DETAILS AND EFFICIENCY

## F.1   GENPRM TRAINING DATA FORMAT

The GenPRM is trained via Supervised Fine-Tuning (SFT) using standard (`prompt, completion`) pairs. Each training example consists of:

1. A natural language OR problem instance.

2. A full step-by-step modeling trajectory generated by the policy model (typically containing 9 steps: problem description, sets, variables, objective, constraints, full model, and code).

3. Task instructions requiring GenPRM to judge each step and provide a concise explanation.

**Prompt Format.** The input prompt aggregates the problem context and the student's trajectory into a single long-context message. The exact template used in our experiments is provided below:

---

**Prompt**

You are a committee of world-class Operations Research (OR) experts.
You will review a student's step-by-step modeling process and provide, for each step, (1) a correctness verdict and (2) a short explanation.
You must follow this output format exactly:
For each step i = 1, ..., 9, output:
STEP_i: [CORRECT/INCORRECT]
EXPLANATION_i: [1-3 sentences explaining your decision]
# CONTEXT
## Problem Description
<problem>
[Full natural-language description of the OR instance]
(e.g., production planning, knapsack, scheduling, network design, ...)
</problem>
## Student's step-by-step trajectory
<step>STEP_1: [Problem restatement and informal formulation] </step>
<step>STEP_2: [Sets and parameters] </step>
<step>STEP_3: [Decision variables] </step>
...
<step>STEP_9: [Final mathematical model or solver code] </step>

---

# TASK
Now, based on the above problem and trajectory, provide:
- For each step i = 1,...,9, a verdict "CORRECT" or "INCORRECT";
- For each step, a short natural-language explanation.
Remember to follow the exact output format specified above.

**Illustrative Example.** To make the above schema more concrete, we provide a shortened example of a GenPRM training instance for a simple knapsack problem.

## Example (prompt excerpt)

**System Instruction:**
You are a committee of world-class Operations Research (OR) experts. You will review a student's step-by-step modeling process and provide, for each step, (1) a correctness verdict and (2) a short explanation. You must follow this output format exactly: for each step $i$, output `STEP_i: CORRECT/INCORRECT` and `EXPLANATION_i: [1--3 sentences]`.

## Problem Description
<problem>
A factory produces two products, A and B. Each unit of A consumes 2 hours of machine time and 1 unit of material, while each unit of B consumes 1 hour of machine time and 2 units of material. The factory has at most 8 hours of machine time and 8 units of material per day. Profit is $3 for each unit of A and $4 for each unit of B. Formulate a linear program that maximizes daily profit.
</problem>

## Student's step-by-step trajectory
<step>
STEP_1: Restate the problem and clarify that the goal is to maximize total daily profit subject to resource constraints on machine time and material.
</step>

<step>
STEP_2: Sets and parameters: let the planning horizon be one day; define available machine time $H = 8$ hours and material $M = 8$ units.
</step>

<step>
STEP_3: Decision variables: $x_A$ = units of product A to produce; $x_B$ = units of product B to produce.
</step>

```
[... remaining steps omitted for brevity ...]
```

## Example (completion excerpt)

```
STEP_1: CORRECT
EXPLANATION_1:  The  student  correctly  restates  the  objective  (maximizing  daily
profit) and notes that the problem is constrained by machine time and material,
which aligns with the original description.

STEP_2: CORRECT
EXPLANATION_2:  The  parameters  H  and  M  are  properly  defined  and  match  the
available resources in the problem statement.

STEP_3: CORRECT
EXPLANATION_3:  The  decision  variables  xA  and  xB  are  clearly  defined  as  daily
production quantities, which is appropriate for this LP formulation.

[... subsequent steps and their explanations omitted ...]
```

**Completion Format.** The target completion follows the fixed schema required by the system instruction. During the self-evolving loop, we use the GenPRM itself to generate these completions.

---
Format

STEP_1: CORRECT / INCORRECT EXPLANATION_1:
[1–3 sentences explaining the verdict]
STEP_2: CORRECT / INCORRECT EXPLANATION_2: [...]
...
STEP_9: CORRECT / INCORRECT EXPLANATION_9: [...]

---

**Solver-Consistent Filtering.** To ensure the quality of these training targets without human annotation, we employ a **solver-consistent filtering** strategy. We only retain training samples where the GenPRM's overall verdict aligns with the external solver's ground truth. Specifically:

- If the solver returns a valid optimal solution, we keep trajectories where GenPRM marks all steps as CORRECT.

- If the solver indicates failure (infeasibility or syntax error), we keep trajectories where GenPRM correctly identifies at least one step as INCORRECT.

This mechanism ensures that the SFT process is anchored by objective solver verification.

## F.2 CONSTRUCTION OF THE SELF-EVOLVING DATASET

The dataset used for the co-evolution loop is derived from the warm-up phase. We generate a large pool of synthetic OR problems and solve each multiple times using the initial policy $\pi_0$ with majority voting.

- **Easy Problems:** Instances where majority voting consistently yields the correct answer are considered "solved" and are excluded from the active training pool to improve efficiency.

- **Challenging Pool:** We identify approximately 13,000 challenging instances where the model either fails completely or shows inconsistency (mixed success rates). These form the shared 13k-problem pool used for the iterative co-evolution loop (Figure 3).

## F.3 ITERATION SCHEDULE AND STOPPING CRITERIA

In each self-evolving iteration, the current policy generates new trajectories on the 13k-problem pool. These are evaluated to construct preference pairs for W-DPO (Policy update) and solver-consistent samples for SFT (GenPRM update).

We perform a total of **three self-evolving iterations** after the warm-up stage. As illustrated in our results (Figure 4 in the main text), the average Pass@1 accuracy improves consistently over the first three iterations. We observed that the performance gain between the third and a potential fourth iteration drops below 1%, indicating saturation. Consequently, we terminate the process and select the checkpoint at the end of the third iteration as the final StepORLM.

## F.4 COMPUTATIONAL COST AND TRAINING EFFICIENCY

We conduct our training on a single node with $8 \times$ NVIDIA H100 80GB GPUs. The computational cost for each stage is detailed as follows:

- **Warm-up SFT:** The initial supervised fine-tuning on the 50K curated dataset (2 epochs) requires approximately **64 GPU-hours**.

- **Self-Evolution Loop:** Each iteration involves trajectory collection, W-DPO policy updates, and GenPRM refinement. The three iterations incur a cumulative cost comparable to the warm-up stage. In total, the full StepORLM pipeline (Policy + GenPRM) requires approximately $10^2$ **GPU-hours**.

This computational footprint is significantly smaller than pre-training or fine-tuning larger generalist models (e.g., 70B parameters), making our framework highly efficient.

