# OpenReview forum: "StepORLM: A Self-Evolving Framework With Generative Process Supervision For Operations Research Language Models"
_ICLR.cc/2026/Conference — ICLR 2026 Poster_

### Official Review · Reviewer_Bj5d · 2025-10-30

**Soundness:** 3
**Presentation:** 3
**Contribution:** 3
**Rating:** 6
**Confidence:** 5

**Summary:**

This paper proposes StepORLM, a self-evolving framework for modeling OR. It aims to address the credit assignment problem, where flawed reasoning might lead to a correct outcome , and the limitations of myopic process supervision that overlooks step interdependencies. The framework's core is a co-evolutionary loop between a policy model and a GenPRM. This loop uses dual feedback from an external solver for outcome verification and the GenPRM for holistic process evaluation. The paper reports that its resulting model achieves state-of-the-art results across six benchmarks and that the GenPRM can be used as a process verifier to improve the inference performance of other LLMs.

**Strengths:**

**Significant Adaptation of Process Supervision:** The paper’s primary strength is its effective adaptation of process-based supervision to the OR domain. By proposing a GenPRM to evaluate entire reasoning trajectories holistically , it aims to mitigate the known limitations of simple outcome-based rewards (the credit assignment problem) and myopic step-wise checks.

**Well-Motivated Co-evolutionary Design:** The paper proposes simultaneous refinement of policy and GenPRM, which is conceptually sound and avoids a static critic. Table 3 shows that freezing GenPRM reduces performance by 3.7% on average, validating the co-evolution mechanism.

**Strong Empirical Results: ** The framework's quality is supported by SOTA performance across six benchmarks. The 8B-parameter StepORLM is shown to outperform vastly larger generalist models and other specialized baselines.

**Weaknesses:**

**Unclear Attribution Due to Ambiguities in the Data Pipeline:** The paper's SOTA claims are complicated by ambiguities in its data strategy. The 50K-sample SFT dataset is significantly larger than baselines like ORLM (30K) , and Figure 4 shows this SFT-only model is already exceptionally strong . This makes it difficult to attribute the final SOTA performance to the co-evolutionary method (Stage 2) rather than just the high-quality initial data (Stage 1). This ambiguity extends to the DPO phase. Section 3.3.1 is unclear whether the DPO phase re-uses these static 50K instances or regenerates new ones , hindering reproducibility.


 **Omission of Relevant Concurrent Work:** The literature review overlooks highly relevant concurrent work that addresses the same problem of intermediate-step validation. Specifically, Step-Opt[1] is directly relevant as it proposes a "Stepwise Validation Mechanism" with explicit checkers for variables, constraints, and programs . This structured, explicit validation is a conceptually similar (though different in approach) solution to the same problem that StepORLM's learned GenPRM aims to solve. Additionally, Lima et al.[2] on verifiable synthetic data is also relevant and should be discussed .

**Backbone Choice Obscures Method Effects:** The paper's SOTA claims are ambiguous. StepORLM is built exclusively on Qwen3-8B , while principal baselines (like the 8B ORLM) rely on different 8B architectures (LLaMA-3-8B). Without cross-backbone controls (e.g., training StepORLM on the baseline's backbone), the comparison fails to isolate the method’s contribution from the backbone’s.

[1] Wu Y, Zhang Y, Wu Y, et al. Step-Opt: Boosting Optimization Modeling in LLMs through Iterative Data Synthesis and Structured Validation[J]. arXiv preprint arXiv:2506.17637, 2025.

[2] Lima V, Phan D T, Kalagnanam J, et al. Toward a trustworthy optimization modeling agent via verifiable synthetic data generation[J]. arXiv preprint arXiv:2508.03117, 2025.

**Questions:**

1. The Data Pipeline: The paper must clarify whether the DPO phase re-uses the static 50K SFT instances or dynamically regenerates new ones. The SFT-only model's results must also be provided to quantify the actual gain from the co-evolutionary method.

2. Omitted Work: The authors must discuss the highly relevant concurrent works by Wu et al. (2025) and Lima et al. (2025)  and situate their novelty accordingly.

3. Backbone Choice: The SOTA claim is confounded by the Qwen3-8B backbone. The authors need to provide experimental results on a common backbone (e.g., LLaMA-3-8B) to enable a fair comparison against baselines like ORLM.

---

> ### Author Response · Authors · 2025-11-24
> **Response to Reviewer Bj5d - Part 1**
>
> We thank the reviewer for the careful and constructive assessment, and for recognizing the adaptation of process supervision to OR, the co-evolutionary design, and the strong empirical results. Below we address the concerns about the data pipeline, concurrent work, and backbone choice.
>
> ## 1. Response to W1 & Q1: Data pipeline, attribution, and SFT vs co-evolution
>
> > *W1: “The 50K-sample SFT dataset is significantly larger than baselines like ORLM (30K)… Figure 4 shows this SFT-only model is already exceptionally strong. This makes it difficult to attribute the final SOTA performance to the co-evolutionary method… Section 3.3.1 is unclear whether the DPO phase re-uses these static 50K instances or regenerates new ones.”*
> > _Q1: “...The SFT-only model's results must also be provided to quantify the actual gain from the co-evolutionary method.”_
>
> We thank the reviewer for raising these points. We would like to clarify (i) the data source and training pipeline for two stages, (ii) the high quality of our curated warm-up SFT data, and (iii) the ablated performance gain from two stages.
>
> ### 1.1 Data Source & Training Pipeline for Stage 1 & 2
>
> Our training pipeline consists of two stages (warm-up SFT and self-evolution):
>
> - **Stage 1 (warm-up SFT).**  We first train the policy model on a curated 50K SFT dataset, which consists of both synthesized OR modeling tasks and OR instruction-style data. Crucially, our data construction methodology represents a significant advancement over the 30k OR-Instruct dataset used in ORLM [1]. With superior scale and quality, as evidenced by our subsequent experiments, we consider the curation of this high-quality warm-up set to be a distinct contribution of our work.
>
> - **Stage 2 (self-evolution via W-DPO + GenPRM).**  We then select **a hard subset of 13K problems from the 50K SFT pool**, consisting of instances where the warm-up policy with 5-way majority voting does not reach unanimous correctness (e.g., vote patterns such as 3:2 or 3:1:1). This 13K hard set is used in the self-evolving iteration loop. In each iteration:
>     1. The current policy generates trajectories on this 13K hard set.
>     2. The external solver verifies the outcomes, and GenPRM evaluates the process.
>     3. We construct preference pairs (winner/loser trajectories) from these newly generated trajectories and train the policy with W-DPO.
>     4. We then use solver-consistent judged trajectories to further fine-tune GenPRM.
>
> Importantly, in Stage 2, the W-DPO preference pairs are constructed from **the solution trajectories newly generated by the current policy on the 13K hard set**, rather than re-using the static 50K SFT instances. We will revise Section 3 to make this data flow explicit and provide a concise pseudo-code description in the appendix to improve reproducibility.
>
> ### 1.2 High Quality of Our Curated Warm-up SFT Data
>
> We conducted experiments to compare the quality of our `warm-up SFT data` against the `OR-Instruct dataset` in ORLM [1]. Although the ORLM paper reports using 30K training samples, only 3K are officially released. To eliminate the impact of data scale/size, we randomly sample a subset of 3K instances from our `warm-up SFT data`. We then performed SFT on the same backbone (LLaMA-3-8B-Instruct) for both datasets, keeping the data size and all other hyperparameters identical.
>
> | Model                                      | NL4OPT | MAMO EasyLP | MAMO ComplexLP | NLP4LP | CompOR | IndOR | ReSocratic |       Avg       |
> | :----------------------------------------- | :----: | :---------: | :------------: | :----: | :----: | :---: | :--------: | :-------------: |
> | LLaMA-3-8B-Instruct + OR-Instruct (3K SFT) |  80.8  |    85.5     |      50.5      |  80.9  |  27.8  | 31.0  |    69.7    |      60.9       |
> | LLaMA-3-8B-Instruct + Warm-up (3K SFT)     |  **89.2**  |    **90.0**     |      **55.0**      |  **87.1** |  **33.3** | **40.5**  |    **72.2**    | **66.8** |
>
> Using the same 3K training samples, our warm-up data yields a **+5.9 point average improvement** over OR-Instruct. This confirms that our performance gains stem primarily from our superior data construction methodology and resulting data quality, rather than merely the larger scale of our full dataset (50k vs. 30k).
>
> We will summarize this controlled comparison in the experimental section and provide full table numbers in the appendix.

---

> > ### Author Response · Authors · 2025-11-24
> > **Response to Reviewer Bj5d - Part 2**
> >
> > ### 1.3 Ablated Performance Gain from Stage 1 & 2
> >
> > While the improvements in each iteration of self-evolving training in Figure 4 may appear modest, the cumulative gain after three iterations of alternating training is substantial. This result validates the core contribution of our proposed framework: the policy model and GenPRM must be trained alternately to mutually reinforce each other, a process that cannot be achieved in a single round.
> >
> > The tables below respectively show the average Pass@1 improvement from self-evolution (Stage 2) over the SFT-only model (Stage 1) when using Qwen3-8B and LLaMA-3-8B-Instruct as the backbone.
> >
> > **Table**: Breakdown Performance of Stage 1 & 2 on Qwen3-8B
> > | Training Stage                      |  NL4OPT  | MAMO EasyLP | MAMO ComplexLP |  NLP4LP  |  CompOR   |   IndOR   | ReSocratic |    Avg    |
> > | :------------------------------- | :------: | :---------: | :------------: | :------: | :-------: | :-------: | :--------: | :-------: |
> > | SFT only                  |   91.1   |    95.4     |      64.9      |   94.9   |   44.4    |   54.8    |    80.1    |   75.1    |
> > | SFT + self-evol.            |   97.7   |    97.2     |      79.3      |   97.8   |   55.6    |   59.5    |    82.6    |   81.4    |
> > | **Improvement over SFT**         | **+6.6** |  **+1.8**   |   **+14.4**    | **+2.9** | **+11.2** | **+4.7**  |  **+2.5**  | **+6.3**  |
> >
> >
> > **Table**: Breakdown Performance of Stage 1 & 2 on LLaMA-3-8B-Instruct
> > | Training Stage                      |  NL4OPT  | MAMO EasyLP | MAMO ComplexLP |  NLP4LP  |  CompOR   |   IndOR   | ReSocratic |    Avg    |
> > | :------------------------------- | :------: | :---------: | :------------: | :------: | :-------: | :-------: | :--------: | :-------: |
> > | SFT only       |   88.3   |    86.2     |      69.4      |   87.6   |   27.8    |   47.6    |    79.2    |   69.4    |
> > | SFT + self-evol. |   97.2   |    97.8     |      84.7      |   93.8   |   55.6    |   61.9    |    84.6    |   82.2    |
> > | **Improvement over SFT**         | **+8.9** |  **+11.6**  |   **+15.3**    | **+6.2** | **+27.8** | **+14.3** |  **+5.4**  | **+12.8** |
> >
> > Our proposed self-evolving training framework can bring an average gain of **+6.3%** on Qwen-8B and **+12.8%** on LLaMA-3-8B-Instruct. The gains are particularly pronounced on the more challenging benchmarks (e.g., +14.4% and +15.3% on ComplexLP), which are precisely the tasks that motivated our self-evolving design.
> >
> > We will highlight these results in Section 4.3 so that the additional effect of the co-evolutionary stage is clearly distinguished from that of the warm-up SFT.
> >
> > ## 2. Response to W2 & Q2: Concurrent work on step-wise validation and verifiable synthetic data
> >
> > > *W2: “The literature review overlooks highly relevant concurrent work… Step-Opt … Lima et al.  … The authors must discuss these works and situate their novelty accordingly.”*
> >
> > We thank the reviewer for pointing out these concurrent works [2, 3], both of which are highly relevant to our setting. At a high level, Step-Opt [2] and Lima et al. [3] both focus on improving the **data side** through iterative synthetic data generation and verifiable pipelines. In contrast, our StepORLM mainly focuses on the **model side** by proposing a co-evolving framework based on process supervision. Detailed discussions are as follows:
> >
> > - **Step-Opt [2]** proposes an iterative data synthesis framework with a stepwise validation mechanism based on explicit checkers for variables, constraints, and programs. Its main goal is to construct high-quality synthetic datasets using rule-based or hand-crafted validators that enforce modeling structure in a structured, step-by-step manner.
> >
> > - **Lima et al. [3]** introduce a pipeline for generating verifiable synthetic data for optimization modeling agents. The pipeline starts from structured symbolic representations and ensures solver-verifiable correctness at each stage. The work focuses on a data generation and verification pipeline that guarantees the correctness of the synthetic corpus used to train optimization agents.
> >
> > The core contribution of StepORLM lies in its co-evolving framework based on process supervision. We believe that by leveraging the data construction schemes from the two methods [2, 3] mentioned above to further enhance data quality, the performance of StepORLM could be boosted even further.
> >
> > We will explicitly add these works to the related work section, clearly mark them as concurrent, and emphasize that they are **complementary rather than overlapping** with StepORLM.

---

> > > ### Author Response · Authors · 2025-11-24
> > > **Response to Reviewer Bj5d - Part 3**
> > >
> > > ## 3. Response to W3 & Q3: Ablation Study w.r.t. Data, Training Method, and LLM Backbone
> > >
> > > > *W3: “StepORLM is built exclusively on Qwen3-8B, while principal baselines (like the 8B ORLM) rely on different 8B architectures (LLaMA-3-8B)… The authors need to provide experimental results on a common backbone (e.g., LLaMA-3-8B) to enable a fair comparison.”*
> > >
> > > We appreciate this important point.  Our initial experiments used Qwen3-8B as the backbone for StepORLM, whereas the ORLM baseline is the official checkpoint released by its authors, which is built on an 8B LLaMA-3 variant. Below we clarify and investigate the performance from three aspects: *data*, *training method*, and *LLM backbone*.
> > >
> > > #### 3.1 Data Aspect: Controlled SFT experiment
> > >
> > > We have already reported the controlled SFT experiment in previous ```1.2 High Quality of Our Curated Warm-up SFT Data```. We copy the content below for your convenience.
> > >
> > > We conducted experiments to compare the quality of our `warm-up SFT data` against the `OR-Instruct dataset` in ORLM [1]. Although the ORLM paper reports using 30K training samples, only 3K are officially released. To eliminate the impact of data scale/size, we randomly sample a subset of 3K instances from our `warm-up SFT data`. We then performed SFT on the same backbone (LLaMA-3-8B-Instruct) for both datasets, keeping the data size and all other hyperparameters identical.
> > >
> > > | Model                                      |  NL4OPT  | MAMO EasyLP | MAMO ComplexLP |  NLP4LP  |  CompOR  |  IndOR   | ReSocratic |   Avg    |
> > > | :----------------------------------------- | :------: | :---------: | :------------: | :------: | :------: | :------: | :--------: | :------: |
> > > | LLaMA-3-8B-Instruct + OR-Instruct (3K SFT) |   80.8   |    85.5     |      50.5      |   80.9   |   27.8   |   31.0   |    69.7    |   60.9   |
> > > | LLaMA-3-8B-Instruct + Warm-up (3K SFT)     | **89.2** |  **90.0**   |    **55.0**    | **87.1** | **33.3** | **40.5** |  **72.2**  | **66.8** |
> > >
> > > Using the same 3K training samples, our warm-up data yields a **+5.9 point average improvement** over OR-Instruct. This confirms that our performance gains stem primarily from our superior data construction methodology and resulting data quality, rather than merely the larger scale of our full dataset (50k vs. 30k).
> > >
> > > #### 3.2 Training Method: Full StepORLM Pipeline on LLaMA-3-8B-Instruct
> > >
> > > We run the full StepORLM training pipeline on LLaMA-3-8B-Instruct, which is the same as ORLM [1]. The decomposed performance evolution across each phase is as follows:
> > >
> > > | Training Phase                                 |  NL4OPT  | MAMO EasyLP | MAMO ComplexLP |  NLP4LP  |  CompOR  |  IndOR   | ReSocratic |   Avg    |
> > > | :--------------------------------------------- | :------: | :---------: | :------------: | :------: | :------: | :------: | :--------: | :------: |
> > > | ORLM Checkpoint (LLaMA-3-8B-Instruct backbone) |   73.8   |    90.4     |      59.5      |   76.4   |   50.0   |   42.9   |    61.8    |   65.0   |
> > > | LLaMA-3-8B-Instruct                            |   19.2   |    18.9     |      14.4      |   18.5   |   11.1   |   7.1    |    23.8    |   16.1   |
> > > | + Warm-up SFT                                  |   88.3   |    86.2     |      69.4      |   87.6   |   27.8   |   47.6   |    79.2    |   69.4   |
> > > | + Self-evol. Iter1                       |   90.1   |    88.4     |      78.4      |   89.3   |   33.3   |   59.5   |    80.6    |   74.2   |
> > > | + Self-evol. Iter2                       |   93.9   |    93.4     |      81.1      |   91.0   |   33.3   |   61.9   |    83.6    |   76.9   |
> > > | + Self-evol. Iter3 (StepORLM)                  | **97.2** |  **97.8**   |    **84.7**    | **93.8** | **55.6** | **61.9** |  **84.6**  | **82.2** |
> > > | + Inference Scaling (StepORLM+GenPRM)          | **97.7** |  **98.0**   |    **87.4**    | **96.6** | **61.1** | **66.7** |  **87.3**  | **85.0** |
> > >
> > > On the same LLaMA-3-8B backbone, our pipeline consistently improves performance from the base model to the warm-up SFT stage (**+53.3%**), then further across three self-evolving iterations (**+12.8%**), and finally through GenPRM-based inference scaling (**+2.8%**). Such a pattern mirrors what we observe on Qwen3-8B (**+17.6%** after SFT, **+9.2%** after 3 iterations, and **+4.2%** after inference scaling). This demonstrates the effectiveness of our proposed training framework.
> > >
> > > #### 3.3 Compatibility across LLM Backbones
> > >
> > > Combining the results on Qwen3-8B (reported in Table 1 of the paper) and the new results above on LLaMA-3-8B-Instruct, we can draw the conclusion that our proposed co-evolving training framework is robustly effective and compatible with different LLM backbones.
> > >
> > > #### **Reference**
> > >
> > > [1] Orlm: A customizable framework in training large models for automated optimization modeling
> > >
> > > [2] Step-Opt: Boosting Optimization Modeling in LLMs through Iterative Data Synthesis and Structured Validation
> > >
> > > [3] Toward a trustworthy optimization modeling agent via verifiable synthetic data generation

---

### Official Review · Reviewer_vMGd · 2025-10-31

**Soundness:** 2
**Presentation:** 1
**Contribution:** 2
**Rating:** 4
**Confidence:** 4

**Summary:**

The paper presents StepORLM, a novel self-evolving framework for LLM-based Operation Research (OR), in which a policy model and a generative process reward model (GenPRM) iteratively improve each other. The framework employs a dual feedback mechanism, combining outcome-based verification from an external solver with process-level evaluation from the GenPRM. The policy model is trained using this combined signal via weighted DPO, while the GenPRM is subsequently refined using new training data generated by the latest policy model. Experimental results demonstrate that the resulting 8B StepORLM model outperforms existing state-of-the-art models across six OR benchmarks, showing the effectiveness of the proposed approach.

**Strengths:**

1. The related work section is clear and reasonable.

2. The proposed self-evolving framework is reasonable and novel to me.

3. The experimental results are strong, demonstrating the effectiveness of the proposed method.

**Weaknesses:**

## Major Concerns

1. Unsupported Motivation (Myopic PRM). While the credit assignment problem is intuitive and makes sense, the myopia PRM problem appears weak to me since many existing PRMs generate process-level supervision signals based on the holistic model output. The authors need to provide stronger evidence or clearer citations to support their claim and better differentiate their work from existing PRMs.

2. There is an apparent disconnect between the paper's claim of using "generative process supervision" and the described implementation. Specifically, according to Algorithm 1 and Equation (1), the detailed, step-by-step process-level correctness scores are seemingly collapsed into a single, global weight $w_{\tau_{w},\tau_{l}}$ for the weighted DPO. This suggests that the fine-grained, process-level feedback is lost and not directly utilized to supervise the model's optimization, which contradicts the paper's central claims.

3. Lack of Clarity on GenPRM Refinement: The methodology for refining the GenPRM is not described in sufficient detail. Key information about the training process is missing, particularly the exact format of the training data.

4. The experimental comparisons in Table 1 are incomplete, making it difficult to fully assess the performance of StepORLM. The missing models include the base model Qwen3-8B, SOTA commercial reasoning models (e.g., OpenAI o-series, Gemini 2.5) and SOTA open-source reasoning models (e.g., DeepSeek-R1, Kimi-K2).

5. Important implementation details are missing, including the number of self-evolving iteration, the prompts for process evaluation and inference scaling experiments, etc.

## Minor Issues

1. L98, “Extensive experiments on seven benchmarks” -> “Extensive experiments on six benchmarks”

2. L298, “ComplexORXiao et al. (2024)”

**Questions:**

While the proposed method demonstrates promising performance on OR tasks, the paper in its current form suffers from several major weaknesses concerning its core motivation, methodological clarity, and experimental rigor. The overall presentation is not up to the ICLR standard. In order to raise my rating, I would like the authors to address the different points in the major concerns.

---

> ### Author Response · Authors · 2025-11-25
> **Response to Reviewer vMGd - Part 1**
>
> We thank Reviewer vMGd for the careful and constructive review. We appreciate that you find the related work section clear, the self-evolving framework reasonable and novel, and the experimental results strong. We would like to address your concerns as follows.
>
> ## 1. Response to Major Concern 1: “Myopic PRM” Motivation
>
> > *Major Concern 1: “Unsupported Motivation (Myopic PRM)… many existing PRMs generate process-level supervision signals based on the holistic model output. The authors need to provide stronger evidence or clearer citations to support their claim and better differentiate their work from existing PRMs.”*
>
> We would like to first clarify the relationship of our work and existing PRMs, and then provide a case study to further support our claim of "Myopic PRM".
>
> ### 1.1 Relation with Existing PRMs and LLM4OR works
>
> We appreciate the opportunity to clarify this. We acknowledge that generative PRMs utilizing holistic outputs exist in the general domain [1, 2]. The existence of these general methods does not diminish our contribution; rather, our novelty lies in identifying and establishing the **design principles for reward mechanisms** specifically in the **solver-aware OR setting**, and proposing a self-evolving framework based on these principles.
>
> Before our work, reward designs for OR modeling with LLMs mostly fall into two categories:
>
> - **Outcome Reward Models (SIRL [3], LLMOPT [4]).** These approaches rely on outcome-only rewards from the solver as RL signals. They capture feasibility and optimality of final solutions but provide very sparse credit assignment to intermediate modeling decisions. In the OR setting, we observe that this can mislead the model to learn brittle modeling patterns that satisfy the solver occasionally but do not correspond to robust reasoning (see also Appendix C in our paper).
>
> - **Discriminative PRMs (Autoformulation [5]).** These methods use discriminative PRMs as non-holistic step-level verifiers and combine them with *inference-time search* (for example, tree search). In OR, however, the objective, variables and constraints are tightly coupled, and they must eventually be implemented consistently in solver code. Verifying each step in isolation is often brittle or infeasible, because many steps only make sense in the context of the full trajectory. Step-level search can also incur substantial computational overhead, especially when many locally plausible steps are globally invalid.
>
> In contrast, our work argues that OR modeling benefits from a **solver-aware generative PRM** that reasons at the trajectory level. We are **the first** to put forward this reward design principle and propose a self-evolving framework tailored for the solver-aware OR setting.
> This specific adaptation to the solver-aware OR context is a significant departure from both existing OR baselines (which use outcome/discriminative rewards) and general PRMs (which lack solver-aware feedback mechanisms).
>
> ### 1.2 Clarification on "Myopic PRM" & Case Study
>
> The term "myopic PRM" in our paper is intended to describe **discriminative, step-wise PRMs** that score each step largely in isolation (as stated in Line 46). However, in OR, variables, indices and constraints interact across distant steps, and the eventual solver code has to implement all of them consistently. A step-wise scorer can therefore reward locally plausible steps that lead to globally infeasible or unimplementable models. We will clarify in the revised paper that this critique is not meant to deny the existence of recent **generative, holistic PRMs** in other domains.
>
> To make the above motivation concrete, we include a compact Traveling Salesman Problem (TSP) case study in the revised appendix. We summarize the key idea below.
>
> In TSP modeling there are two standard formulations:
> - DFJ formulation is **impractical for solver code**, since it uses only binary variables $x_{ij}$ but requires exponential subtour constraints;
> - MTZ formulation is **practical for solver code**, since it introduces auxiliary continuous variables $u_i$ and only needs polynomially many static constraints.
>
> Now given a specific TSP instance in our case study, the policy generates a solution trajectory, which, at step 3, defines only $x_{ij}$ and implicitly heads toward the DFJ formulation (impractical for solver code). Then, discriminative and generative PRMs can behave differently as follows.

---

> ### Author Response · Authors · 2025-11-25
> **Response to Reviewer vMGd - Part 2**
>
> **1.2.1 Discriminative PRM: local correctness and reward hacking**
>
> A discriminative PRM **scores each step largely in isolation**. Hence, it labels step 3 of the generated trajectory as ```CORRECT``` since the variable choice is mathematically valid for TSP, so the policy is rewarded for a seemingly simpler formulation.
> However, when generating code later, it becomes **practically infeasible** to implement the exponentially many subtour constraints in a single static Python script, so the resulting model is broken or severely relaxed.
>
> This leads to a form of **reward hacking** for the policy. The policy finds a way to receive a local reward for a mathematically elegant but practically unusable modeling path, because the PRM *does not connect early decisions with downstream solver failure*.
>
> **1.2.2 Generative PRM: global reasoning and consistency**
>
> Our GenPRM, in contrast, **evaluates the entire trajectory** with a chain-of-thought style critique that explicitly reasons about later steps and solver behavior. For the same step 3 (defining only $x_{ij}$), GenPRM reads the full trajectory, including the failed code in the final step, and labels this decision as ```INCORRECT```.
>
> By using global, trajectory-level reasoning to align early steps with downstream feasibility, GenPRM can prevent the reward hacking of policy. It would penalize the policy to refine the trajectory with MTZ formulation and finally produces executable, solver-feasible code, *instead of rewarding a locally correct but globally wrong step*.
>
> **1.2.3 Reliability with Dual-Feedback Mechanism**
>
> Although GenPRM prevents reward hacking, it should be further strengthened by solver outcomes. In our experiment, the reliability could be generally guaranteed based on our proposed **dual-feedback mechanism** (GenPRM + Solver Outcome), i.e., we do not observe any systematic reward hacking that degrades performance.
>
> ## 2. Response to Major Concern 2: Generative Process Supervision & Scalar Weighted DPO
>
> > *Major Concern 2: “There is an apparent disconnect between the paper's claim of using ‘generative process supervision’ and the described implementation… step-by-step process-level correctness scores are seemingly collapsed into a single, global weight… fine-grained feedback is lost and not directly utilized to supervise the model's optimization.”*
>
> This concern touches a central design choice in our framework. Our clarification and experience can be summarized along three points.
>
> ### 2.1 Direct process-level objectives were tried but proved unstable
>
> In early experiments, we did try to use GenPRM's step-level signals as *direct process supervision* for the policy, for example, via step-level DPO-style losses or other per-step preference objectives. In practice, these variants were difficult to train. We observed non-convergence or significantly worse final accuracy, with very noisy gradients and strong sensitivity to small alignment errors at the step level.
>
> The current W-DPO formulation aggregates the process supervision signals into a scalar weight, which is a practical compromise among three goals: (i) stable optimization, (ii) effective use of GenPRM's process information, and (iii) strong end-task performance. With this design, StepORLM already achieves state-of-the-art average Pass@1 on six OR benchmarks and outperforms strong closed models such as Gemini-2.5-Pro and OpenAI-o3, as reported `4. Response to Major Concern 4`.
>
> ### 2.2 How generative process supervision is still used in our framework
>
> Even though the policy training loss aggregates the process supervision signals into a scalar weight (i.e., W-DPO), this weight is derived from GenPRM's step-level process evaluation. Moreover, GenPRM's **generative process supervision** plays a central role in two ways.
>
> - **GenPRM training.**  GenPRM itself is trained on full trajectory critiques. In the self-evolving stage, we let GenPRM generate step-wise judgments and explanations on trajectories for which GenPRM’s verdict is consistent with the solver outcome. We then treat these generated critiques as SFT targets. This directly optimizes GenPRM to produce high-quality process supervision on trajectories that are solver-consistent.
>
> - **Inference-time verification and scaling.**  At inference time, we treat GenPRM as a process verifier. For each problem, we sample multiple trajectories and let GenPRM judge the entire reasoning process. Its trajectory-level evaluation is used to accept or reject candidate solutions. This improves the final accuracy of StepORLM and also of external OR LLMs (e.g., ORLM [6]) when we apply GenPRM to them.
>
> Thus, generative process supervision is used both to refine GenPRM itself and to define the weights in the W-DPO objective for the policy, rather than being discarded.

---

> > ### Author Response · Authors · 2025-11-25
> > **Response to Reviewer vMGd - Part 3**
> >
> > ### 2.3 Outlook and sharing negative results
> >
> > We believe there is significant potential to design more sophisticated objectives that leverage step-level process signals more directly while maintaining training stability and strong performance, for instance through improved reinforcement learning methods or refined credit assignment schemes. We consider this a pivotal direction for future work building upon StepORLM.
> >
> > Furthermore, we plan to include the unsuccessful process supervision variants that we explored but ultimately did not adopt in our open-source code release. We appreciate the reviewer's feedback and insight, which makes us realize that these negative results can also offer valuable lessons for the LLM4OR community. We hope that sharing them will stimulate further research into robust process-level objectives.
> >
> > ## 3. Response to Major Concerns 3 & 5: Clarity on GenPRM refinement and implementation details
> >
> > > *Major Concern 3: “The methodology for refining the GenPRM is not described in sufficient detail. Key information about the training process is missing, particularly the exact format of the training data.”*
> > > *Major Concern 5: “Important implementation details are missing, including the number of self-evolving iterations, the prompts for process evaluation and inference scaling experiments, etc.”*
> >
> > We thank the reviewer for highlighting these issues. In the following, we would like to provide (i) overview of data source and training pipeline for StepORLM, (ii) the exact training data format for GenPRM, and  (iii) key implementation details and hyperparameter configurations.
> >
> > ### 3.1 Overview of data source and training pipeline
> >
> > Our training pipeline consists of two stages: a warm-up SFT stage and a self-evolving stage.
> >
> > - **Stage 1 (warm-up SFT).**  We first train the policy model on a curated 50K SFT dataset, which consists of both synthesized OR modeling tasks and OR instruction-style data. The detailed data format and prompt template in warm-up SFT stage are already provided in Appendix C.
> >
> > - **Stage 2 (self-evolution via W-DPO and GenPRM).**  We then select a **hard subset of 13K problems from the 50K SFT pool**, consisting of instances where the warm-up policy with 5-way majority voting does not reach unanimous correctness (e.g., vote patterns such as 3:2 or 3:1:1). This 13K hard set is used in the self-evolving iteration loop. In each iteration:
> >     1. The current policy generates trajectories on this 13K hard set.
> >     2. The external solver verifies the outcomes, and GenPRM evaluates the process.
> >     3. We construct preference pairs (winner/loser trajectories) from these newly generated trajectories and train the policy with W-DPO.
> >     4. We then use solver-consistent judged trajectories to further fine-tune GenPRM.
> >
> > Importantly, in Stage 2, the W-DPO preference pairs are constructed from **the solution trajectories newly generated by the current policy on the 13K hard set**, rather than re-using the static 50K SFT instances.
> >
> > ### 3.2 Training data format for GenPRM
> >
> > Since we have already provided the data template for policy training in Appendix C, here, we give the detailed training data format and prompt for GenPRM.
> >
> > GenPRM is refined with standard instruction-tuning style **(prompt, completion)** pairs. Each training example contains: (i) a natural-language OR problem instance; (ii) a full step-by-step modeling trajectory produced by the current strategy model (problem description, sets/parameters, decision variables, objective, constraints, full model, code, etc.); and (iii) task instructions that ask GenPRM to judge **each step** and provide a short explanation.
> >
> > Thus the prompt is a single long-context message:
> >
> > ```txt
> > You are a committee of world-class Operations Research (OR) experts.
> > You will review a student's step-by-step modeling process and provide,
> > for each step, (1) a correctness verdict and (2) a short explanation.
> >
> > You must follow this output format exactly:
> > For each step i, output:
> > STEP_i: [CORRECT/INCORRECT]
> > EXPLANATION_i: [1-3 sentences explaining your decision]
> >
> > # CONTEXT
> >
> > ## Problem Description
> > <problem>
> > [Full natural-language description of the OR instance]
> > </problem>
> >
> > ## Student's step-by-step trajectory
> > <step>
> > STEP_1: [Problem restatement and informal formulation]
> > </step>
> >
> > <step>
> > STEP_2: [Sets and parameters]
> > </step>
> >
> > <step>
> > STEP_3: [Decision variables]
> > </step>
> >
> > ...
> >
> > <step>
> > STEP_9: [Final mathematical model or solver code]
> > </step>
> >
> > # TASK
> > Now, based on the above problem and trajectory, provide:
> > - For each step, a verdict "CORRECT" or "INCORRECT";
> > - For each step, a short natural-language explanation.
> > Remember to follow the exact output format specified above.
> > ```
> >
> > The completion then strictly follows a fixed schema:
> >
> > ```txt
> > STEP_1: CORRECT / INCORRECT
> > EXPLANATION: [...]
> >
> > STEP_2: CORRECT / INCORRECT
> > EXPLANATION: [...]
> >
> > ...
> >
> > STEP_9: CORRECT / INCORRECT
> > EXPLANATION: [...]
> > ```

---

> > > ### Author Response · Authors · 2025-11-25
> > > **Response to Reviewer vMGd - Part 4**
> > >
> > > During self-evolution, we use GenPRM itself to generate these completions, but only on high-confidence instances where GenPRM's overall verdict is consistent with the external solver’s outcome. For such trajectories, we treat GenPRM’s own evaluation output, including the holistic critique and step-wise judgments, as targets and continue to fine-tune GenPRM on these prompt–completion pairs. This yields a self-training loop for GenPRM that is anchored by solver agreement. We will add this schema and one illustrative example to the appendix so that the GenPRM training format is completely transparent.
> > >
> > > ### 3.3 Implementation Details and Hyperparameter Settings
> > >
> > > In addition to the implementation details in Section 4.1, here, we will provide more details and hyperparameter settings for SFT, self-evolution, and inference scaling.
> > >
> > > - **Warm-up SFT stage.** We perform full-parameter fine-tuning of the policy model. We introduce special tokens such as `<step>` and `</step>` to mark step boundaries and use a maximum sequence length of 16,384 tokens. We train with AdamW, a learning rate of $1\times 10^{-5}$, two epochs, per-device batch size 2, and gradient accumulation of 8. We use a cosine learning rate schedule with a 10% warmup ratio and apply gradient clipping with maximum gradient norm 1.0. Training is run in bf16 with FlashAttention 2 enabled.
> > >
> > > - **W-DPO stage.**  For W-DPO, we use a fixed temperature parameter $\beta = 0.1$ in the DPO loss. We use AdamW with learning rate $5\times 10^{-7}$, a global batch size of 64, and a cosine schedule with 10% warmup.
> > >
> > > - **Inference scaling with GenPRM.** For inference scaling, we use a fixed sampling budget of $k = 4$ trajectories per problem. The final GenPRM obtained after three self-evolving iterations serves as a verifier. For each problem, we repeatedly sample candidate trajectories under the same decoding configuration and let GenPRM check their process quality. We adopt a trajectory once it passes GenPRM’s verification or we reach the sampling limit $k$. In all inference-scaling experiments, we use sampling temperature 0.9, top-p value 0.95, and maximum generation length 16,384 tokens.
> > >
> > > We hope that all the clarifications above (i.e., data source and training pipeline, GenPRM training data format, and the hyperparameter settings) could address your major concerns 3 & 5, and make the self-evolving framework easier to reproduce for readers.
> > >
> > > ## 4. Response to Major Concern 4: Completeness of baselines in Table 1
> > >
> > > > _MC4: “The experimental comparisons in Table 1 are incomplete… missing the base model Qwen3-8B, SOTA commercial reasoning models (OpenAI o-series, Gemini 2.5) and SOTA open-source reasoning models (DeepSeek-R1, Kimi-K2).”_
> > >
> > > We fully agree that these baselines are important for a comprehensive comparison. In response to this and other reviewers’ suggestions, we have updated Table 1 to include all of the requested models:
> > > - **Base model**: Qwen3-8B;
> > > - **Commercial reasoning models**: OpenAI o3, Gemini-2.5-Pro;
> > > - **Open-source reasoning models**: DeepSeek-R1, Kimi-K2.
> > >
> > > ### Updated Table 1: Pass@1 accuracy (%) on six OR benchmarks
> > >
> > > Abbreviations: CompOR = ComplexOR, IndOR = IndustryOR, Avg = macro-average. All agentic methods use GPT-4o as the backbone.
> > >
> > > |Model|Params|NL4OPT|MAMO-EasyLP|MAMO-ComplexLP|NLP4LP|CompOR|IndOR|ReSocratic|Avg|
> > > |:--|:-:|:-:|:-:|:-:|:-:|:-:|:-:|:-:|:-:|
> > > |Zero-shot LLMs||||||||||
> > > |OpenAI o3|Closed|78.4|93.9|63.1|93.8|72.2|76.2|84.4|80.3|
> > > |Gemini-2.5-Pro|Closed|82.6|87.9|52.3|94.9|66.7|78.6|89.6|78.9|
> > > |GPT-4o|Closed|61.2|70.3|57.7|73.6|42.9|38.1|48.4|56.0|
> > > |Kimi-K2|1T|77.9|93.4|55.9|84.3|61.1|59.5|81.9|73.4|
> > > |DeepSeek-R1|671B|77.5|90.3|59.5|87.6|66.7|59.5|83.9|75.0|
> > > |DeepSeek-V3|671B|79.8|95.2|53.2|92.1|55.6|66.7|85.1|75.4|
> > > |Qwen2.5-72B-Inst|72B|78.9|95.8|44.1|88.2|50.0|57.1|81.1|70.7|
> > > |Qwen3-32B|32B|77.5|92.3|46.9|93.8|50.0|61.9|85.1|72.5|
> > > |Qwen3-8B|8B|63.8|73.6|45.0|58.4|27.8|52.4|61.0|54.6|
> > > |Fine-tuned LLMs||||||||||
> > > |ORLM|8B|73.8|90.4|59.5|76.4|50.0|42.9|61.8|65.0|
> > > |LLMOPT|14B|75.1|83.5|67.6|86.0|22.2|52.4|73.2|65.7|
> > > |OptMATH|32B|95.9*|89.9*|54.1*|–|–|–|–|–|
> > > |StepORLM|8B|97.7|97.2|79.3|97.8|55.6|59.5|82.6|81.4|
> > > |Agentic methods||||||||||
> > > |OptiMUS-v0.3|Closed|76.2|78.0|46.8|88.8|46.8|45.2|87.6|67.1|
> > > |CoT|Closed|62.2|49.5|42.3|74.7|39.2|40.5|43.6|50.3|
> > > |CoE|Closed|66.7|94.4|50.6|87.4|57.1|31.2|71.2|65.5|
> > > |CAFA|Closed|68.1|71.2|44.5|50.0|46.4|41.1|40.1|51.6|
> > > |StepORLM+GenPRM|8B+8B|97.2|97.8|87.4|98.9|61.1|61.9|94.6|85.6|
> > >
> > > As shown in the table above, these newly included large, strong generalist reasoning models perform pretty well, especially OpenAI-o3 and Gemini-2.5-Pro. However, our proposed **StepORLM (8B)** and **StepORLM+GenPRM (8B+8B)** still achieve the **highest** average Pass@1 accuracy across all six OR benchmarks, despite using only 8B parameters for the policy backbone.

---

> > > > ### Author Response · Authors · 2025-11-25
> > > > **Response to Reviewer vMGd - Part 5**
> > > >
> > > > We believe that the updated table better supports the claim that StepORLM is competitive with, and often surpasses, state-of-the-art generalist and specialized reasoning models in the LLM4OR setting, validating the effectiveness of our proposed co-evolving training framework.
> > > >
> > > > ## 5. Minor issues
> > > >
> > > > > _L98, “Extensive experiments on seven benchmarks” -> “six benchmarks”_
> > > > > _L298, “ComplexORXiao et al. (2024)”_
> > > >
> > > > We thank the reviewer for pointing out these typos. We will correct L98 to “six benchmarks” and add the missing space in “ComplexOR Xiao et al. (2024)” in the revised manuscript.
> > > >
> > > > Once again, we sincerely thank Reviewer vMGd for the detailed comments and for the positive assessment of our framework and results. Your questions led us to clarify the scope of the “myopic PRM” discussion, to better explain how GenPRM is trained and used within the self-evolving loop, and to strengthen both the experimental comparisons and the implementation details. We believe that the revisions described above will substantially improve the clarity and rigor of the paper, and we hope that the updated version will satisfactorily address your concerns.
> > > >
> > > > ## References
> > > >
> > > > [1] R-PRM: Reasoning-Driven Process Reward Modeling
> > > >
> > > > [2] GenPRM: Scaling Test-Time Compute of Process Reward Models via Generative Reasoning
> > > >
> > > > [3] Solver-Informed RL: Grounding Large Language Models for Authentic Optimization Modeling
> > > >
> > > > [4] LLMOPT: Learning to Define and Solve General Optimization Problems from Scratch
> > > >
> > > > [5] Autoformulation of Mathematical Optimization Models Using LLMs
> > > >
> > > > [6] ORLM: A customizable framework in training large models for automated optimization modeling

---

> > > > > ### Comment · Reviewer_vMGd · 2025-11-26
> > > > >
> > > > > Thank you for the clarifications. My concerns have been addressed by the point-by-point rebuttals and I'm willing to increase my score to reflect the hard-works being done.

---

### Official Review · Reviewer_hi6J · 2025-10-31

**Soundness:** 3
**Presentation:** 3
**Contribution:** 2
**Rating:** 4
**Confidence:** 3

**Summary:**

StepORLM introduces a self-evolving 8B-parameter framework that couples a policy model with a generative process reward model (GenPRM) to train LLMs for operations-research tasks. By iteratively distilling dual feedback—solver-verified outcomes and holistic trajectory-level critiques—into weighted DPO updates, the system attains new state-of-the-art accuracy on six benchmarks, while the co-evolved GenPRM further boosts inference-time scaling for both its own policy and external OR models.

**Strengths:**

The proposed method is simple yet effective, achieving performance improvements.

**Weaknesses:**

GenPRM and Self-Improve are not novel concepts; they have been explored in prior research, eg. [1][2][3][4].

[1] R-PRM: Reasoning-Driven Process Reward Modeling

[2] GenPRM: Scaling Test-Time Compute of Process Reward Models via Generative Reasoning

[3] rStar-Math: Small LLMs Can Master Math Reasoning with Self-Evolved Deep Thinking

[4] Inference-Time Scaling for Generalist Reward Modeling

**Questions:**

Why is there such a large performance gap between LLMOPT (origin) and LLMOPT (reproduce)?

What model is the CoT on line 344 based on?

The StepORLM performance in Tables 2 and 3 is inconsistent with that in Table 1.

---

> ### Author Response · Authors · 2025-11-24
> **Response to Reviewer hi6J - Part1**
>
> We thank the reviewer for acknowledging our method as "simple yet effective" and recognizing the SOTA performance improvements. We value the opportunity to clarify our contributions relative to prior work and address the specific questions regarding our experimental setup.
>
> ## 1. Response to W1: Novelty of GenPRM and self-improvement vs [1–4]
>
> > _W1: “GenPRM and Self-Improve are not novel concepts; they have been explored in prior research, e.g. [1]–[4].”_
>
> We would like to clarify that our contribution is not to “invent” generative PRMs or self-improvement [1–4]. Our contribution lies in **how** these ideas are instantiated, adapted, and combined in the OR + solver context, which have not been explored before:
>
> - **OR + solver–centric generative PRM.** Prior works [1–4] mainly target math Q&A or general reasoning, where the supervision is an answer token or verbal correctness. In StepORLM, GenPRM is designed for OR modeling with an external solver: outcomes are feasibility/optimality certificates, and intermediate steps are OR primitives (sets, variables, constraints, solver code) that must be semantically consistent with the solver.
>
> - **Dual-feedback W-DPO combining outcome and process signals.** Instead of using GenPRM only for reranking or as a scalar reward, we distill two signals into a single weighted DPO objective: (i) solver-verified outcome correctness to define preferences, and (ii) GenPRM’s step-correctness ratio to define pair-specific weights. Such an OR-oriented dual-feedback signal largely contributes to the superior performance of StepORLM.
>
> - **Explicit co-evolution of policy and generative PRM.** Whereas prior self-evolving methods (e.g., rStar-Math) evolve a single policy, StepORLM co-evolves two models: the policy is updated via W-DPO using solver+GenPRM feedback, and GenPRM is continuously refined by SFT on solver-verified trajectories produced by the current policy. This closes a two-way feedback loop between the solver, policy, and GenPRM.
>
> - **GenPRM as a reusable OR verifier.** We show that the learned GenPRM can be deployed as a process verifier at inference time not only for StepORLM itself (StepORLM+GenPRM), but also for external OR models (e.g., ORLM), yielding substantial gains without retraining those models.
>
> We will revise the related work and method sections to clearly position StepORLM relative to [1–4] along these dimensions, emphasizing that our framework is based on the generative supervision idea in previous works, but instantiated with a dual-feedback objective and a two-model co-evolution design tailored to OR with solver feedback.
>
> ## 2. Response to Q1: LLMOPT (origin) vs LLMOPT (reproduce)
>
> > *Q1: “Why is there such a large performance gap between LLMOPT (origin) and LLMOPT (reproduce)?”*
>
> We thank the reviewer for drawing our attention to this discrepancy. In fact, ever since completing our initial experiments we had been aware of this gap and had been actively trying to resolve it. After contacting the LLMOPT authors, through joint discussions we identified that the main cause was an incomplete match to their original test-time environment, and we worked together to fix this issue.
>
> With the corrected environment, LLMOPT’s performance on our cleaned benchmarks improves substantially. Under the aligned setup, LLMOPT achieves the following accuracies on our cleaned benchmark splits:
>
> | Model                          | NL4OPT | MAMO EasyLP | MAMO ComplexLP | NLP4LP | CompOR | IndOR | ReSocratic |
> | ------------------------------ | :----: | :---------: | :------------: | :----: | :----: | :---: | :--------: |
> | LLMOPT (origin)                |  80.3  |    89.5     |      44.1      |  73.4  |  35.3  | 29.0  |    53.8    |
> | LLMOPT (reproduce, before fix) |  49.3  |    36.3     |      25.2      |  43.3  |  16.7  | 40.5  |    39.5    |
> | LLMOPT (reproduce, after fix)  |  75.1  |    83.5     |      67.6      |  86.0  |  22.2  | 52.4  |    73.2    |
>
> We have also updated this report in the manuscript and the subsequent updated Table 1. After this correction, **StepORLM and StepORLM+GenPRM still outperform LLMOPT on all benchmarks** (see the updated Table 1 in the revision), so our main conclusions remain unchanged. At the same time, the corrected results more accurately reflect the strength of LLMOPT under a fair and well-aligned setting.
>
> We sincerely thank the LLMOPT authors for their help in diagnosing the environment mismatch, and we appreciate the reviewer for the kind and constructive reminder. We believe this clarification and update will help the community better compare methods in the emerging LLM4OR area and foster constructive collaboration between different research groups.

---

> ### Author Response · Authors · 2025-11-24
> **Response to Reviewer hi6J - Part2**
>
> ## 3. Response to Q2: Model used for the CoT example (line 344)
>
> > *Q2: “What model is the CoT on line 344 based on?”*
>
> We thank the reviewer for the question. We would like to clarify that this information is already stated in the paper (**line 316**). Thus, the CoT trajectory shown on line 344 is generated by the **GPT-4o-based CoT baseline**. To avoid any possible ambiguity for future readers, we will revise the manuscript to restate in the caption of the table reporting agentic baselines that all agentic methods, including CoT, use GPT-4o as their backbone, making this detail more immediately visible.
>
> ## 4. Response to Q3: StepORLM performance inconsistency
>
> > *Q3: “The StepORLM performance in Tables 2 and 3 is inconsistent with that in Table 1.”*
>
> We sincerely appreciate your crucial reminder. We apologize for the inconsistency between Table 1 and Tables 2–3: Table 1 mistakenly reported an **earlier checkpoint** of StepORLM. We have now updated Table 1 to use the **same final checkpoint** as in Tables 2–3, and at the same time incorporated several **stronger baselines** (e.g., OpenAI o3, Gemini-2.5-Pro, Kimi-K2, DeepSeek-R1, Qwen3-8B) requested by other reviewers. The updated results are shown below; StepORLM and StepORLM+GenPRM still achieve the highest average Pass@1 accuracy among all compared methods.
> ### Updated Table 1: Pass@1 accuracy (%) on six OR benchmarks
>
> Abbreviations: CompOR = ComplexOR, IndOR = IndustryOR, Avg = macro-average. All agentic methods use GPT-4o as the backbone.
>
> | Model               | Params |  NL4OPT  | MAMO-EasyLP | MAMO-ComplexLP |  NLP4LP  |  CompOR  |  IndOR   | ReSocratic |   Avg    |
> | :------------------ | :----: | :------: | :---------: | :------------: | :------: | :------: | :------: | :--------: | :------: |
> | **Zero-shot LLMs**  |        |          |             |                |          |          |          |            |          |
> | OpenAI o3           | Closed |   78.4   |    93.9     |      63.1      |   93.8   |   72.2   |   76.2   |    84.4    |   80.3   |
> | Gemini-2.5-Pro      | Closed |   82.6   |    87.9     |      52.3      |   94.9   |   66.7   |   78.6   |    89.6    |   78.9   |
> | GPT-4o              | Closed |   61.2   |    70.3     |      57.7      |   73.6   |   42.9   |   38.1   |    48.4    |   56.0   |
> | Kimi-K2             |   1T   |   77.9   |    93.4     |      55.9      |   84.3   |   61.1   |   59.5   |    81.9    |   73.4   |
> | DeepSeek-R1         |  671B  |   77.5   |    90.3     |      59.5      |   87.6   |   66.7   |   59.5   |    83.9    |   75.0   |
> | DeepSeek-V3         |  671B  |   79.8   |    95.2     |      53.2      |   92.1   |   55.6   |   66.7   |    85.1    |   75.4   |
> | Qwen2.5-72B-Inst    |  72B   |   78.9   |    95.8     |      44.1      |   88.2   |   50.0   |   57.1   |    81.1    |   70.7   |
> | Qwen3-32B           |  32B   |   77.5   |    92.3     |      46.9      |   93.8   |   50.0   |   61.9   |    85.1    |   72.5   |
> | Qwen3-8B            |   8B   |   63.8   |    73.6     |      45.0      |   58.4   |   27.8   |   52.4   |    61.0    |   54.6   |
> | **Fine-tuned LLMs** |        |          |             |                |          |          |          |            |          |
> | ORLM                |   8B   |   73.8   |    90.4     |      59.5      |   76.4   |   50.0   |   42.9   |    61.8    |   65.0   |
> | LLMOPT              |  14B   |   75.1   |    83.5     |      67.6      |   86.0   |   22.2   |   52.4   |    73.2    |   65.7   |
> | OptMATH             |  32B   |  95.9*   |    89.9*    |     54.1*      |    –     |    –     |    –     |     –      |    –     |
> | **StepORLM**        |   8B   | **97.7** |  **97.2**   |    **79.3**    | **97.8** | **55.6** | **59.5** |  **82.6**  | **81.4** |
> | **Agentic methods** |        |          |             |                |          |          |          |            |          |
> | OptiMUS-v0.3        | Closed |   76.2   |    78.0     |      46.8      |   88.8   |   46.8   |   45.2   |    87.6    |   67.1   |
> | CoT                 | Closed |   62.2   |    49.5     |      42.3      |   74.7   |   39.2   |   40.5   |    43.6    |   50.3   |
> | CoE                 | Closed |   66.7   |    94.4     |      50.6      |   87.4   |   57.1   |   31.2   |    71.2    |   65.5   |
> | CAFA                | Closed |   68.1   |    71.2     |      44.5      |   50.0   |   46.4   |   41.1   |    40.1    |   51.6   |
> | **StepORLM+GenPRM** | 8B+8B  | **97.2** |  **97.8**   |    **87.4**    | **98.9** | **61.1** | **61.9** |  **94.6**  | **85.6** |

---

### Official Review · Reviewer_Yvzp · 2025-11-02

**Soundness:** 2
**Presentation:** 3
**Contribution:** 2
**Rating:** 6
**Confidence:** 4

**Summary:**

This paper introduces StepORLM, a novel framework for training large language models to solve operations research (OR) problems more reliably. The key innovation is generative process supervision through a Generative Process Reward Model (GenPRM) that holistically evaluates full reasoning trajectories, rather than step-by-step discriminative scoring. StepORLM optimizes a policy model through a dual-feedback loop: (1) outcome verification from an external solver ensures correctness of final results, and (2) process evaluation from GenPRM assesses reasoning quality. Training proceeds via Weighted Direct Preference Optimization (W-DPO) to align the policy with high-quality reasoning. Meanwhile, it also optimizes GenPRM during the process. Experiments on six benchmarks show StepORLM (8B parameters) outperforming larger models like GPT-4o, Qwen3-32B, and specialized ORLM baselines, achieving state-of-the-art Pass@1 accuracy. The paper also shows GenPRM’s transferability as a universal process verifier for inference-time scaling.

**Strengths:**

1. Introduces generative process supervision, a step beyond discriminative PRMs, enabling trajectory-level reasoning assessment and solving the credit assignment problem.
2. The co-evolution between policy and GenPRM provides a mutually reinforcing refinement process that avoids overfitting.
3. Empirical results are strong with demonstrated consistent SOTA performance across six diverse OR benchmarks, outperforming vastly larger generalist and agentic systems.

**Weaknesses:**

1. Most benchmarks are still synthetic or adapted from prior academic datasets; stronger demonstration on industrial-scale, more real-world tasks would enhance credibility.
2. The framework assumes deterministic, verifiable outcomes from OR solvers; applicability to open-ended or ill-posed reasoning tasks remains uncertain.
3. The dual-model co-evolution loop may be expensive to scale, and the paper provides limited discussion of training efficiency.
4. While numerical gains are clear, qualitative examples of how GenPRM critiques or refines reasoning are sparse. For example, any reward hacking examples from the PRM would be helpful.

**Questions:**

1. Can this method be used in other domains outside OR?
2. Do you observe any reward hacking using PRM? Is Generative PRM always reliable?
3. How good is the generalization of this algorithms across dataset and domain?

---

> ### Author Response · Authors · 2025-11-24
> **Response to Reviewer Yvzp - Part1**
>
> We thank the reviewer for the thoughtful and positive assessment, and for highlighting (i) the move beyond discriminative PRMs to generative process supervision, (ii) the co-evolutionary design between policy and GenPRM, and (iii) the strong empirical results. Below we address the concerns regarding real-world applicability, domain generality, training efficiency, and qualitative evidence.
>
> ## 1. Response to W1: Real-world vs. synthetic benchmarks
>
> > _W1: "Most benchmarks are still synthetic or adapted from prior academic datasets; stronger demonstration on industrial-scale, more real-world tasks would enhance credibility.”_
>
> We agree that more industrial-scale, proprietary datasets would further strengthen the work. At the same time, we would like to clarify that our evaluation already includes a genuinely industrial benchmark:
>
> - **IndustryOR.** As described in Appendix B.5, IndustryOR comprises **42 verified real-world industrial scenarios** drawn from **manufacturing, logistics, and finance**. These problems involve realistic, noisy textual descriptions and cover a mix of **linear, integer, mixed-integer, and nonlinear programming** tasks. They are non-trivial: for example, GPT-4o only achieves 38.1% Pass@1 accuracy on this benchmark.
>
> On the IndustryOR dataset, StepORLM achieves 59.5% Pass@1, outperforming GPT-4o (38.1%) and DeepSeek-V3 (55.6%). This suggests that our framework is not only effective on synthetic or academic-style problems, but also on realistic industrial cases.
>
> We fully agree that evaluating on larger-scale, proprietary industrial workloads would be valuable future work, for example full production planning systems or logistics platforms within a single company. However, such data are typically not publicly available and are hard to release. Our current goal is to demonstrate strong performance on publicly available OR benchmarks, including those derived from real industrial scenarios, to ensure reproducibility and fair comparison.
>
> ## 2. Response to W2 & Q1 & Q3 (part): Applicability beyond OR and dependency on deterministic solvers
>
> > _W2: “The framework assumes deterministic, verifiable outcomes from OR solvers; applicability to open-ended or ill-posed reasoning tasks remains uncertain.”_
> > _Q1: “Can this method be used in other domains outside OR?”_
> > _Q3: “How good is the generalization of this algorithms across dataset and domain?”_
>
> Our design **intentionally targets domains with strong, programmatic verifiers**, and OR is a canonical example because high-quality solvers are available. We discuss the design philosophy and applicable domain as follows:
>
> - **Design premise.** StepORLM relies on an external solver to provide deterministic outcome verification (feasibility and optimality), which is combined with GenPRM’s process judgments. This is not a limitation of the framework, but rather a deliberate choice of domain where we can address credit assignment in a principled way without noisy outcome labels.
>
> - **Potentially applicable domains.** Conceptually, the same structure (external verifier + generative process supervisor) can be applied to other reasoning domains with programmatic checkers, such as:
>     - Code generation (compiler + unit tests as verifiers)
>     - Formal theorem proving (proof assistants like Lean/Isabelle as verifiers)
>     - Other formal verification tasks.
>
>     In these settings, the solver’s role is replaced by a task-specific checker, while GenPRM still enforces process-level consistency.
>
> - **Open-ended domains.** For truly open-ended or ill-posed tasks (e.g., free-form dialogue, subjective QA), there is no deterministic solver. In principle, one could replace the solver with a strong reward model or human preference labels, but this brings substantial challenges (reward hacking, bias, inconsistency). We therefore **do not claim** applicability to such domains in the current work; extending StepORLM to these settings is an interesting but non-trivial direction for future work.
>
> Within OR, however, the framework shows **strong generalization across diverse datasets** (see also our `5. Response to Q3 (continued)` below): from textbook-style LPs (NL4OPT, MAMO) to more open-ended industrial descriptions (IndustryOR) and hardness-oriented benchmarks (ComplexOR, ReSocratic). We will clarify in the introduction and conclusion that the current scope is **structured reasoning tasks with reliable programmatic verifiers**, and position extensions beyond OR as future work.

---

> ### Author Response · Authors · 2025-11-24
> **Response to Reviewer Yvzp - Part2**
>
> ## 3. Response to W3: Training efficiency and co-evolution cost
>
> > _W3: “The dual-model co-evolution loop may be expensive to scale, and the paper provides limited discussion of training efficiency.”_
>
> We appreciate this comment and agree that the current version under-specifies the compute cost. In the revised version, we will provide a concise description of the training footprint.
>
> Concretely:
>
> - **Warm-up SFT.** On 8× H100 GPUs, the warm-up SFT stage over our 50K examples (2 epochs) takes about 8 hours wall-clock, i.e., roughly ***64 GPU-hours***.
>
> - **Self-evolution (3 iterations).** The three self-evolving iterations (policy rollouts on the 13K challenging-problem pool, W-DPO updates for the policy, and SFT refinement for GenPRM) add a ***similar amount of compute*** as the warm-up SFT stage. In total, the full StepORLM pipeline on a single backbone (policy + GenPRM) requires on the order of ***10² GPU-hours***.
>
> - **Relative cost.** This overhead is modest compared to pre-training or fine-tuning much larger models (e.g., 70B–600B-scale generalist models), yet it allows an 8B policy to consistently outperform such larger models across six OR benchmarks.
>
> We will explicitly add these details (training stages, number of iterations, and order-of-magnitude GPU-hours) to the experimental section, so that readers can better understand the efficiency and overhead of the co-evolution loop. We are open to releasing more details if you have any further questions or suggestions.
>
> ## 4. Response to W4 & Q2: Reward hacking and GenPRM reliability (TSP case study)
>
> > _W4: “While numerical gains are clear, qualitative examples of how GenPRM critiques or refines reasoning are sparse. For example, any reward hacking examples from the PRM would be helpful.”_
> > _Q2: “Do you observe any reward hacking using PRM? Is Generative PRM always reliable?”_
>
> We agree that more qualitative insight is valuable, and we appreciate the opportunity to illustrate how GenPRM mitigates reward hacking. Below we give a compact Traveling Salesman Problem (TSP) **case study**. The detailed case study is provided in the revised paper; below we only give the key points.
>
> In TSP modeling there are two standard formulations:
> - DFJ formulation is **impractical for solver code**, since it uses only binary variables $x_{ij}$ but requires exponential subtour constraints;
> - MTZ formulation is **practical for solver code**, since it introduces auxiliary continuous variables $u_i$ and only needs polynomially many static constraints.
>
> Now given a specific TSP instance in our case study, the policy generates a solution trajectory, which, at step 3, defines only $x_{ij}$ and implicitly heads toward the DFJ formulation (impractical for solver code). Then, discriminative and generative PRM can behave differently as follows.
>
> ### 4.1 Discriminative PRM: local correctness and reward hacking
>
> A discriminative PRM **scores each step largely in isolation**. Hence, it labels the step 3 of the generated trajectory as ```CORRECT``` since the variable choice is mathematically valid for TSP, so the policy is rewarded for a seemingly simpler formulation.
> However, when generating code later, it becomes **practically infeasible** to implement the exponentially many subtour constraints in a single static Python script, so the resulting model is broken or severely relaxed.
>
> This leads to a form of **reward hacking** for the policy. The policy finds a way to receive a local reward for a mathematically elegant but practically unusable modeling path, because the PRM *does not connect early decisions with downstream solver failure*.
>
> ### 4.2 Generative PRM: global reasoning and consistency
>
> Our GenPRM, in contrast, **evaluates the entire trajectory** with a chain-of-thought style critique that explicitly reasons about later steps and solver behavior. For the same step 3 (defining only $x_{ij}$), GenPRM reads the full trajectory, including the failed code in the final step, and labels this decision as ```INCORRECT```.
>
> By using global, trajectory-level reasoning to align early steps with downstream feasibility, GenPRM can prevent the reward hacking of policy. It would penalize the policy to refine the trajectory with MTZ formulation and finally produces executable, solver-feasible code, *instead of rewarding a locally correct but globally wrong step*.
>
> ### 4.3 Reliability with Dual-Feedback Mechanism
>
> Although GenPRM prevents reward hacking, it should be further strengthened by solver outcomes. In our experiment, the reliability could be generally guaranteed based on our proposed **dual-feedback mechanism** (GenPRM + Solver Outcome), i.e., we do not observe any systematic reward hacking that degrades performance.
>
> We will add this TSP case study to the appendix to more clearly show how our GenPRM refines reasoning and mitigates reward hacking compared with discriminative PRMs.

---

> > ### Author Response · Authors · 2025-11-24
> > **Response to Reviewer Yvzp - Part3**
> >
> > ## 5. Response to Q3 (continued): Generalization across datasets and models
> >
> > > _Q3: “How good is the generalization of this algorithms across dataset and domain?”_
> >
> > Within OR, we evaluate StepORLM across **six diverse benchmarks** (NL4OPT, MAMO Easy/Complex, NLP4LP, ComplexOR, IndustryOR, ReSocratic). These cover:
> > - Textbook-style linear programs and modeling tasks
> > - Harder combinatorial problems with complex constraints (ComplexOR, ReSocratic)
> > - Real-world industrial scenarios (IndustryOR).
> >
> > On all six benchmarks, StepORLM consistently improves over the base backbone and prior OR-specific baselines, indicating strong **cross-dataset generalization** within the OR domain.
> >
> > Moreover, we show that **GenPRM generalizes across policy models**. It can be used as a **process verifier for other OR LLMs** (e.g., ORLM) during inference-time scaling, improving their performance without retraining those models. This suggests that GenPRM is generalized and is not tightly overfitted to a single policy’s output distribution.
> >
> > As discussed in `2. Response to W2 & Q1 & Q3 (part)`, extending these generalization results **beyond OR** to other structured domains (code, theorem proving, etc.) is conceptually feasible but remains future work; we deliberately limited our empirical claims to OR tasks with strong solvers to ensure a trustworthy evaluation.
> >
> > Once again, we thank Reviewer Yvzp for the constructive feedback. We will (i) better highlight the real-world nature of IndustryOR, (ii) more clearly state the domain assumptions and potential extensions, (iii) document the compute footprint of the co-evolution loop, and (iv) add qualitative case studies (such as the TSP example) to show how GenPRM mitigates reward hacking and enforces global consistency. We believe these revisions will significantly improve the clarity and impact of the paper.

---

### Author Response · Authors · 2025-12-04
**Summary of Rebuttal**

Dear Chairs,

We sincerely thank the reviewers for their thoughtful feedback. We are encouraged that the reviewers recognize the **novelty** of our generative process supervision, the **soundness** of the co-evolutionary framework, and the **strong SOTA results** of StepORLM.

Crucially, we wish to highlight the status of our rebuttal **prior to the massive data-leak incident (27 Nov around 2:00, AoE)**:
- Our paper achieved ratings of ```6,6,6,4```, with **5.5 on average**.
- Reviewer **vMGd**, who is the only reviewer who had the opportunity to respond, explicitly stated _"My concerns have been addressed..."_ and thereby **raised the rating to 6** before the data-leak incident.
- Due to the early termination of the discussion, the remaining three reviewers (hi6J, Bj5d, Yvzp) did not have the opportunity to participate further. However, we have provided comprehensive new experiments and clarifications that directly resolve all their specific concerns.

Below we provide a concise summary of our rebuttal and clarifications to support your final assessment.

---
## Consensus Among Reviewers
- **Novelty & Significance**: Reviewers **vMGd**, **Yvzp**, and **Bj5d** explicitly highlight that our self-evolving framework and the move from discriminative to generative process supervision are novel and well-motivated for the OR domain. Reviewer **hi6J** also acknowledges the method as "simple yet effective."

- **Strong Empirical Results**: **All reviewers** agree that StepORLM demonstrates strong, consistent SOTA performance across diverse benchmarks, significantly outperforming vastly larger generalist models and specialized baselines.

- **Sound Framework**: Reviewers **Bj5d** and **Yvzp** validate the co-evolutionary design, noting that the simultaneous refinement of Policy and GenPRM effectively avoids overfitting and provides mutual reinforcement.
---
## Summary of Revisions and Clarifications
We address every concern raised by the reviewers with new experiments and revisions:
- **Motivation & Contribution (Yvzp, vMGd)**:
    - **Motivation of "Myopic PRM"**: We add a comprehensive **TSP Case Study in Appendix C** to illustrate "reward hacking" in discriminative PRMs, which tend to favor mathematically valid yet code-impractical formulations (i.e., myopic). In contrast, we demonstrate that our GenPRM enforces global consistency, effectively avoiding this pitfall.
    - **Generalization of GenPRM**: We clarify how GenPRM is incorporated in both training and inference. We further justify that our proposed framework is generalized to other domains with strong, programmatic verifiers.

- **More SOTA Baselines (vMGd)**: We include more SOTA generalist reasoning models (*OpenAI o3, Gemini-2.5-Pro, DeepSeek-R1, Kimi-K2, Qwen3-8B*) as baselines. StepORLM (8B) still **outperforms** these large models on average.

- **Detailed Ablations (Bj5d)**: We conduct additional experiments for breakdown analysis of performance improvement from the following aspects in Appendix D:
    - **Backbone Independence**: We run the full StepORLM pipeline on the **LLaMA-3-8B** backbone (same as the baseline ORLM). The results show a **+12.8%** improvement over SFT, confirming that gains stem from our _method_, not the specific Qwen backbone.
    - **SFT vs. Co-evolution:** We provide controlled ablations showing that while our high-quality SFT data yields a **+5.9%** gain over prior datasets, the self-evolution stage contributes an additional **+6.3% to +12.8%**, verifying the necessity of both stages.
    - **Data Pipeline:** We clarify that the DPO phase uses _newly generated_ trajectories, not static SFT data.

- **More Related Works (Bj5d)**: We discuss more recent related works like Step-Opt, and point out that these works, which focuses on *data side*, are **complementary rather than overlapping** with our StepORLM which focus on *model side*.

- **More Implementation Detail (hi6J, vMGd, Bj5d)**: We provide the exact training data format and prompt templates, as well as all the hyperparameter settings, for the reproducibility of StepORLM in Appendix E and F.

- **Training Cost (Yvzp):** We disclose that the full pipeline requires only **~100 GPU hours**, demonstrating high training efficiency.

- **Real-World Applicability (Yvzp)**: We highlight our performance on IndustryOR (42 real-world industrial scenarios & problems), where StepORLM achieves **59.5%** accuracy compared to GPT-4o's 38.1%.

- **LLMOPT Reproduction (hi6J)**: We contact the original LLMOPT authors, identify an environment mismatch, and correct the results. StepORLM remains superior to the corrected baseline.

We have highlighted all modifications in the revised paper in blue, and we hope the Area Chair will consider our rebuttal and the corresponding clarifications. We believe these substantial additions adequately address the reviewers' concerns and justify the positive consensus formed by reviewers.

Thank you again for your time and effort!

Best regards,

Authors

---

### Meta-Review · Area_Chair_jZi9 · 2026-01-06

**Summary:**

StepORLM introduces a self-evolving framework using Generative Process Supervision (GenPRM) to solve Operations Research (OR) problems. By co-evolving a policy model and a generative reward model through solver-based outcome verification and trajectory-level process evaluation, the 8B model achieves SOTA performance. The reviewers are overall positive. The primary concerns from reviewers are as follows.
- Data vs. Method: Whether gains stemmed from the novel framework or simply a high-quality 50K SFT dataset.
- Real-world Generalization: Dependency on deterministic solvers and performance on non-synthetic industrial tasks.
- Training Cost: The feasibility and expense of scaling the iterative co-evolution loop.
- Reward Hacking: Lack of evidence that GenPRM avoids "myopic" errors common in discriminative PRMs.

The authors' rebuttal addressed the concerns generally (one reviewer increased the score from 4 to 6).
- Ablations: Confirmed that self-evolution contributes an additional 6–13% gain beyond SFT.
- Industrial Proof: Success on the IndustryOR benchmark, outperforming GPT-4o on real-world scenarios.
- Efficiency: The full pipeline requires only ~100 GPU hours.
- Qualitative Evidence: A TSP case study demonstrated GenPRM's ability to ensure global consistency, effectively mitigating reward hacking.

**Reviewer Concerns:**

The authors' addressed the concerns generally. Reviewer vMGd acknowledged the rebuttal and raised the rating to 6

**Reviewer Scores:**

Reviewer vMGd acknowledged the rebuttal and raised the rating to 6.

---

### Decision · Program_Chairs · 2026-01-26

Accept (Poster)